# Expand Neurons, Not Parameters

**Linghao Kong** [* 1]  **Inimai Subramanian** [* 1]  **Yonadav Shavit** [2]  **Micah Adler** [1]  **Dan Alistarh** [3 4]  **Nir Shavit** [1 4]

## Abstract

This work demonstrates how increasing the number of neurons in a network without increasing its total number of non-zero parameters improves performance. We show that this gain corresponds with a decrease in interference between multiple features that would otherwise share the same neurons. On symbolic Boolean tasks, splitting each neuron into sparser sub-neurons with knowledge of the clauses systematically reduces polysemanticity metrics and yields higher task accuracy. Notably, even random splits of neuron weights approximate these gains, indicating that reduced collisions, not precise assignment, are a primary driver. Consistent with the superposition hypothesis, the benefits of this framework grow with increasing interference: when polysemantic load is high, accuracy improvements are the largest. Transferring these insights to more realistic models, including classifiers over CLIP embeddings, convolutional neural networks, and deeper multilayer networks, we find that widening networks while maintaining a constant non-zero parameter count consistently increases accuracy. These results identify an interpretability-grounded mechanism to leverage width against superposition, improving performance without increasing the number of non-zero parameters. Such a direction is well matched to modern accelerators, where memory movement of non-zero parameters, rather than raw compute, is often a dominant bottleneck.

[*]Equal contribution  [1]Massachusetts Institute of Technology [2]Independent Researcher [3]Institute of Science and Technology Austria [4]Red Hat AI. Correspondence to: Linghao Kong <linghao@mit.edu>, Inimai Subramanian <inimai@mit.edu>, Nir Shavit <shanir@mit.edu>.  Code available at https://github.com/Shavit-Lab/Expand-Neurons.

*Proceedings of the 43rd International Conference on Machine Learning*, Seoul, South Korea. PMLR 306, 2026. Copyright 2026 by the author(s).

## 1. Introduction

Understanding the mechanisms behind neural network performance has become increasingly crucial as models grow in complexity, scale, and ubiquity. Despite their widespread adoption, neural networks frequently remain opaque due to polysemantic neurons: individual neurons that simultaneously encode multiple features (Goh et al., 2021; Jermyn et al., 2022; Gurnee et al., 2024; Dreyer et al., 2024), frustrating interpretability. This entanglement induces interference among features that are forced to share the same neuron, degrading performance (Lecomte et al., 2023).

Two complementary lines of work highlight this phenomenon. The superposition hypothesis argues that networks represent more features than they have neurons, leading to interference when distinct features coexist within a single unit, initially through empirical findings in toy models (Olah et al., 2020; Elhage et al., 2022; Liu et al., 2025). Recent theoretical analyses formalize the parameter requirements for representing features and show that features often compete at the level of edges, not just neurons (Adler & Shavit, 2024; Hänni et al., 2024; Adler et al., 2025). Meanwhile, the lottery ticket hypothesis (Frankle & Carbin, 2018) and follow-up work (Chen et al., 2020; 2021; Zhou et al., 2019) suggest that sparse subnetworks with the right connectivity can match or exceed dense performance, again pointing to structure, not density, as the key determinant.

Together, these perspectives motivate a concrete question. If features interfere because they are forced to share neurons, what happens if we keep the total number of non-zero parameters fixed but allocate those edges across more neurons? In this work, we answer this question by demonstrating that increasing the number of neurons while keeping the number of non-zero parameters fixed consistently reduces interference and improves performance in interference-limited regimes.

In Boolean settings, where ground-truth feature structure is known, we show that widening the network without adding parameters reliably reduces feature interference. Gram-matrix block structure becomes sharper, feature capacity increases (Scherlis et al., 2022), and activations become more selective. These geometric changes closely mirror improvements in accuracy, revealing a tight interference-performance relationship. We then test the same principle beyond symbolic tasks, including Boolean models with

learned embeddings, classifier layers with convolutional neural network (CNN) backbones, and CLIP-based (Radford et al., 2021) CIFAR (Krizhevsky et al., 2009) and ImageNet (Deng et al., 2009) classifiers. Across these settings, increasing neurons while keeping parameters fixed systematically reduces interference in high-superposition regimes and improves performance.

To summarize, our contributions are as follows:

1. We show that redistributing a fixed non-zero parameter budget across more neurons reduces feature interference and improves accuracy, emphasizing neuron count as an axis of model performance distinct from parameter count.

2. We provide a theoretical analysis showing that expanding width at fixed parameter count preserves feature coverage while reducing expected collisions, even under random weight partitioning.

3. We validate this framework across symbolic Boolean tasks and real-world vision benchmarks, with gains largest in high-interference regimes, as predicted by the superposition hypothesis.

4. We provide direct mechanistic evidence linking width expansion to reduced polysemanticity: feature capacity increases, cosine similarity decreases, and both strongly correlate with accuracy gains.

The rest of the paper is organized as follows. Section 2 introduces the neuronal expansion procedure and the experimental setup used to vary neuron count while preserving the non-zero parameter budget. Section 3 provides a theoretical analysis of why expansion can preserve feature coverage while reducing collisions in Boolean tasks. Section 4 evaluates expansion in symbolic settings, where ground-truth features allow direct measurement of interference and clause recovery. Section 5 extends the analysis to real-world vision settings, including frozen embeddings and jointly learned representations. Section 6 discusses related work, and Section 7 concludes with limitations and future directions.

Before proceeding, we clarify the scope of our claims. As in prior mechanistic interpretability work (e.g., Elhage et al. 2022), our experiments focus on small models and controlled settings where representational structure can be analyzed directly. Our goal is to characterize a tractable regime in which the relationship between neuron count, feature interference, and performance can be studied mechanistically. Extending to larger architectures, including transformers, remains an important direction for future work.

**Conflict of interest disclosure**
YS was previously employed by OpenAI, which released CLIP (Radford et al., 2021). We use CLIP embeddings.

## 2. Methodology

### 2.1. Classifier architecture

To examine the impact of neuron count under a fixed parameter budget, we begin with a fully connected feedforward architecture with a single hidden layer. Let $\mathbf{x} \in \mathbb{R}^d$ be the input, and let $h$ denote the number of neurons. The classifier maps inputs to output logits via the transformation $\mathbf{z} = \mathbf{W_2} \cdot \text{ReLU}(\mathbf{W_1}\mathbf{x} + \mathbf{b_1}) + \mathbf{b_2}$, where $\mathbf{W_1} \in \mathbb{R}^{h \times d}$ and $\mathbf{W_2} \in \mathbb{R}^{C \times h}$ are the weight matrices of the first and second layers respectively, $\mathbf{b_1} \in \mathbb{R}^h$, $\mathbf{b_2} \in \mathbb{R}^C$ are biases, and $C$ is the number of output classes. In binary classification settings, we apply a final sigmoid activation to the output $\mathbf{z}$ and use binary cross-entropy loss. In multiclass classification, we treat $\mathbf{z}$ as unnormalized logits and apply standard softmax-based cross-entropy loss.

This relatively minimal architecture is intentionally underparameterized in terms of neurons to study the emergence of superposition and interference. For ImageNet-1k, we use a deeper five layer network, constructed in the same way. For our experiments with joint feature learning, we either use a simple embedding layer $\mathbf{E}$ for Boolean tasks or a CNN backbone, both of which are trainable and feed into a classifier. More details in Sections A.7.1 and A.8.

### 2.2. The neuronal expansion procedure

In this work, we use the edge-partitioning transformation as an experimental probe of the interference-performance relationship. We refer to this specific instantiation as Fixed Parameter Expansion (FPE), but our goal is not to introduce a deployment-ready recipe. Rather, FPE is a controlled intervention that varies neuron count and feature collisions at fixed parameter count, allowing us to measure resulting changes in internal geometry and task performance.

Let the baseline dense model have hidden width $h$, input size $d$, and output size $C$, as defined in Section 2.1. For an integer expansion factor $\alpha > 1$, we define the expanded width $h' = \alpha h$, and construct a new weight matrix $\mathbf{W'_1} \in \mathbb{R}^{h' \times d}$ with a binary sparsity mask $\mathbf{M_1} \in \{0, 1\}^{h' \times d}$. For each original neuron $n_i$ with weight vector $\mathbf{w_i}$, we duplicate $\mathbf{w_i}$ across $\alpha$ sub-neurons $n_{\alpha i:\alpha(i+1)}$ in $\mathbf{W'_1}$. The input dimension is partitioned into $\alpha$ disjoint masks $\mathbf{m_{(i_k)}} \in \{0, 1\}^d$ of roughly equal total non-zero parameters, such that $\sum_{k=1}^{\alpha} \mathbf{m_{(i_k)}} = \mathbf{1}_d$, and each mask is applied to one sub-neuron. This ensures no sub-neurons share input features, and the total number of non-zero parameters in $\mathbf{W'_1}$ equals that in $\mathbf{W_1}$. If biases are used, we copy the original bias to all sub-neurons to form $\mathbf{b'_1}$.

To handle the wider hidden layer, we use a "re-sparsification" strategy: We expand $\mathbf{W_2} \in \mathbb{R}^{C \times h}$ to $\mathbf{W'_2} \in \mathbb{R}^{C \times h'}$ by duplicating each original output weight vector $\mathbf{w_j}$ across $\alpha$

sub-features. $\mathbf{b}_2'$ is directly copied from the original. This results in $(\alpha - 1)C$ excess parameters. To preserve parameter count, the smallest-magnitude weights in $\mathbf{W}_1'$ and $\mathbf{W}_2'$ are pruned, and masks $\mathbf{M}_1$ and $\mathbf{M}_2$ are updated accordingly. The masks are not updated following initialization. This process is extended to deeper networks for our ImageNet-1k experiments, and either a simulated embedding layer $\mathbf{E}$ or a CNN backbone is prepended to this classifier for our joint representation learning and classification experiments. More details in Sections A.7.1 and A.8, including experimenting with mask updates.

The masks $\mathbf{m}_{(\mathbf{i_k})}$ are constructed either randomly or using feature-based grouping. In Boolean tasks, clause-aware splitting assigns all inputs from a clause to the same sub-neuron. In vision tasks, we emulate group structure by observing the Gram matrix, as described in Sections A.2, 4.1 and 5. In all experiments other than the illustrative example (Figure 1), the dense classifier is first trained for 25 warmup epochs, during which it closely approaches convergence and learns stable mixed features. We then apply FPE to this near-converged model, expanding the architecture under the same parameter budget. Following this, both the dense and FPE models are fine-tuned for an additional 25 epochs under identical training settings (Section A.3). This procedure allows us to evaluate whether increasing neuron count alone, without increasing parameter count, can reduce superposition interference and improve performance.

## 2.3. Tasks and datasets

### Symbolic reasoning

For precise structural control, we construct a series of classification tasks based on satisfiable monotone read-once Boolean formulas (O'Donnell, 2014), expressed in disjunctive normal form (DNF). Each formula is a disjunction of conjunctive clauses, each of which has exactly four literals. All literals are positive and appear exactly once across the entire formula. For example: $(x_1 \wedge x_2 \wedge x_3 \wedge x_4) \vee (x_5 \wedge x_6 \wedge x_7 \wedge x_8) \vee (x_9 \wedge x_{10} \wedge x_{11} \wedge x_{12}) \cdots$

Each input is a modified binary truth assignment to the full set of variables, and the label indicates whether the assignment satisfies the formula. This formulation creates one distinct, non-overlapping feature per clause, with no interference from variable reuse or negation. By increasing the number of clauses, we increase the number of independent features to be represented, thereby intensifying superposition pressure. Analogously, by reducing the number of hidden neurons in the classifier, we decrease the network's representational capacity. This dual control allows us to induce or relieve superposition either from the feature side, via clause count, or from the neuron side, via layer width, enabling a clear experimental probe of the representational bottleneck. More details in Section A.3.

### Visual classification

We test our framework on FashionMNIST (Xiao et al., 2017), CIFAR-100 (Krizhevsky et al., 2009), ImageNet-100 (constructed via a random selection of 100 ImageNet classes), and ImageNet-1k (Deng et al., 2009). For Fashion-MNIST, we use raw flattened grayscale images. For CIFAR-100 and ImageNet, we evaluate on features extracted using a frozen CLIP ViT-B/16 encoder (Radford et al., 2021), treating these embeddings as fixed inputs to the classifier. Using frozen embeddings decouples our analysis from representation learning dynamics and isolates the effect of expansion under a fixed non-zero parameter budget on classification performance. In addition, we include experiments on raw images from CIFAR-100 using a CNN backbone trained jointly with the classifier, providing evidence that neuronal expansion remains effective when representations and the classifier are learned end-to-end. Additional training details in Sections A.3 and A.7.1.

## 3. Theoretical Analysis

To provide a theoretical justification of how increasing width without changing the total number of non-zero parameters reduces interference, we analyze a network with $r$ neurons trained on a Boolean DNF task of $m$ total literals with $k$ literals per clause. In this setting, we consider clauses and features equivalent. The network's $r$ neurons are duplicated into $\alpha r$ sub-neurons while preserving the total number of non-zero weights. This is achieved by sparsifying each sub-neuron to degree $d = \frac{m}{\alpha}$. We find that this preserves clause coverage with high probability while cutting expected clause collisions by $\approx \alpha^{-(2k-1)}$. Intuitively, coverage is easy because many neurons sample the needed literals, while collisions are rare because all $2k$ inputs of two features must land within the same neuron's sparse support. This controlled analysis provides a motivating framework for our experiments, which we extend to settings with realistic features. The full justification can be found in Section A.5.

We also situate our setup within the feature channel coding hypothesis (Adler et al., 2025), which complements the superposition view of Elhage et al. (2022). In this framework, abstract features are implemented by feature channels, which are sets of neurons that share a characteristic weight-sign pattern. Adler et al. (2025) give explicit constructions and combinatorial packing bounds for such codes, showing that with a fixed number of neurons there is a finite capacity for reliably representing Boolean features, and as more features are crammed into the same set of rows, code overlaps and errors must increase. Our analysis of expanding width is aligned with this view: at fixed non-zero parameter budget, duplicating neurons increases the number of rows available to host feature-channel codes, which reduces expected overlaps or collisions without sacrificing clause coverage.

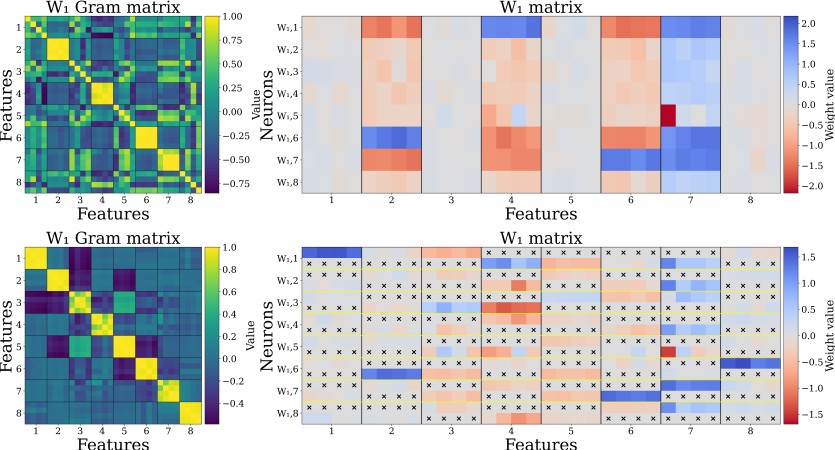

*Figure 1.* More neurons yields less interference between features. On a Boolean task in disjunctive normal form with clauses of four literals each, clear codes are visible for clauses 2, 4, 6, and 7 for the dense model in the first row. For a network with the same number of non-zero parameters but twice as many neurons, clear codes are visible for all clauses in the second row. Black x's represent masked parameters of value 0. Additional details, including definition of codes, in Section 4.1.

This perspective both motivates our focus on weight-level geometry, such as through feature capacity and cosine similarity, and explains why even random splitting should reduce interference. Our goal in this section is not to present a complete theory of neuronal expansion, but to show that our analysis sits within a now partly formalized view of superposition. While the original superposition hypothesis was introduced from empirical and toy-model evidence (Elhage et al., 2022), subsequent work on feature-channel codes (Adler et al., 2025) derives explicit combinatorial capacity bounds for the same kind of overlapping codes. Our collision analysis can be read as a simplified instance of this framework: duplicating neurons at fixed non-zero parameter budget increases the number of rows available for such codes, thereby reducing overlaps.

## 4. Results on Symbolic Tasks

### 4.1. A case study in neuronal expansion

We first provide an illustrative example of how FPE reduces interference. We train the classifier from Section 2.1 on a DNF formula with 8 clauses of 4 distinct literals each, $\vee_{j=1}^{8} \left( \wedge_{i=1}^{4} x_{4(j-1)+i} \right)$, treating each clause as a feature.

We first train a compact model with 8 hidden neurons, then expand it with $\alpha = 2$ using either clause-based or random splitting (Section A.3), and continue training all models, including the dense one, under the same non-zero parameter budget (Section A.3.3). To analyze the learned representations we examine the first-layer Gram matrix $G = \mathbf{W}_1^\top \mathbf{W}_1$ (Yang & Chaudhari, 2025), which summarizes how first-layer weights span feature space: strong on-diagonal blocks indicate well-isolated clause codes, while large off-diagonal values indicate entanglement. With 8 neurons, the dense

model reaches 78.7% test accuracy, compared to 99.4% for clause-split FPE and 88.7% for random-split FPE. Figures 1 and A2 show the corresponding Gram matrices and $\mathbf{W}_1$, with vertical lines marking clause boundaries and horizontal lines grouping sub-neurons.

The dense model's Gram matrix exhibits clear codes for some clauses but weaker, more diffuse structure for others, consistent with feature interference. By codes, we mean the patterns of the signs of the weights in a neuron associated with a particular clause, specifically four positive weights, combined with a negative bias and a positive weight in the subsequent layer. This configuration implements a soft logical AND (Gupta, 1999): the neuron remains inactive and does not fire unless all four literals are present in the boolean formula, thus giving the neuron sufficient strength to overcome the negative bias. Following activation, it contributes positively to the output. As a result, the neuron selectively fires on inputs satisfying the clause. Importantly, Adler et al. (2025) show that the emergence of such codes during training is tightly correlated with improvements in network performance. Clause splitting produces a strongly block-diagonal Gram matrix and more specialized weights, recovering previously entangled clauses while keeping the parameter count fixed. Random splitting yields weaker block structure, yet still improves over the dense baseline and sometimes approaches clause-split performance.

These observations support the superposition hypothesis: the compact model is limited by neurons sharing multiple features, and increasing width under a fixed parameter budget, especially with structured clause-aware splitting, reduces such collisions and improves performance.

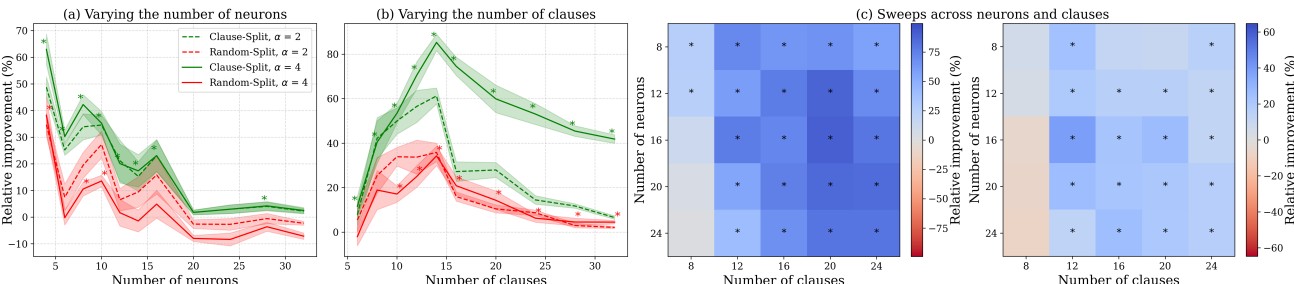

*Figure 2.* Trends of performance under superposition in neurons and clauses. (a) Relative improvement in test accuracy for models of varying hidden dimension on 8 clauses. (b) Relative improvement in test accuracy for models with 8 neurons. y-axis labels are shared for (a) and (b). (c) Heatmap of relative improvement percentage for clause-split FPE models (left) and random-split FPE models (right). Relative improvement is calculated as $\frac{\text{FPE test accuracy} - \text{dense test accuracy}}{\text{dense test accuracy}}$. Error bars indicate one standard error of the mean. * indicates $p < 0.05$ and is shown only for $\alpha = 4$ for clarity. Results collected over five trials.

### 4.2. Scaling interference with neurons and clauses

We now train a classifier on a Boolean DNF formula with clauses of four literals each, while varying the number of neurons and clauses the dense network is trained on. We expand the dense models to an expansion factor $\alpha = \{2, 4\}$, and evaluate the performance of a dense model, a clause-split model, and a random-split model.

**Effect of neuron count**
Increasing the number of neurons leads to diminishing returns in relative improvement for learning 8 clauses (Figure 2a). This is consistent with our expectation: with fewer neurons for the same number of features, superposition-derived interference increases, which neuron splitting via FPE effectively reduces. Clause splitting consistently outperforms random splitting, although the latter still often outperforms the dense baseline.

**Effect of clause count**
As the number of clauses increases for a network with 8 neurons, performance gains from FPE initially rise, reflecting increased feature entanglement in the dense model. Again, clause-split yields higher improvements than random-split. However, beyond $\approx 16$ clauses, the benefit of splitting begins to plateau (Figure 2b). This suggests that a network's capacity to fully disentangle complex features is eventually saturated even with more neurons.

**Scaling behavior**
Figure 2c shows heatmaps of performance improvement, highlighting the joint effects of clause count and neuron count. These results support the interpretation that FPE is most beneficial when feature demand is high relative to available neurons. FPE gains generally increase with clause count and often decrease with neuron count, consistent with larger benefits under higher superposition pressure and smaller benefits under lower pressure, respectively. The joint sweep shows the tradeoff between these effects: improvements persist along the diagonal of Figure 2c when

clause count and neuron count increase together. We also observe a plateau in performance gains for models with 8, 12, and 16 neurons, suggesting that beyond a certain point, even splitting cannot fully disentangle the clause structure. Finally, random splitting provides modest improvements across many configurations. While clause-based splitting consistently performs best, random splitting still improves over the dense baseline in this controlled setting, highlighting the general effectiveness of increasing neuron capacity under a fixed non-zero parameter budget.

**Direct evidence of feature disentanglement**
To empirically show that FPE reduces feature entanglement within the neural network, we measure superposition on the Boolean task using feature capacity analysis (Scherlis et al., 2022) and orthogonality.

For each feature (clause), its capacity $C_i$ is calculated as a measure of interference. The capacity allocated to feature $i$ is defined as: $C_i = \frac{(W_{\cdot,i} \cdot W_{\cdot,i})^2}{\sum_j (W_{\cdot,i} \cdot W_{\cdot,j})^2}$, where $W$ are the activations of the model (which in our Boolean setting is the same as the weight matrix) and $W_{\cdot,i}$ is the activations of the $i$th feature. $C_i$ is the fraction of the dimension allocated to representing feature $i$, with the numerator representing the squared magnitude of feature $i$'s weight vector, and the denominator accounting for the total overlap of feature $i$ with all other features, thus measuring how much of the representational space is uniquely dedicated to feature $i$ versus shared with other features. High capacity is indicative of lower polysemanticity, as neurons are more dedicated to representing single features. We then sum all individual feature capacities to measure overall capacity.

To reduce polysemanticity, neuron weight vectors should also be increasingly orthogonal. The mean cosine similarity is calculated by averaging the cosine similarity between all pairs of distinct neuron weight vectors. Lower similarity is indicative of lower polysemanticity, as features are represented more independently.

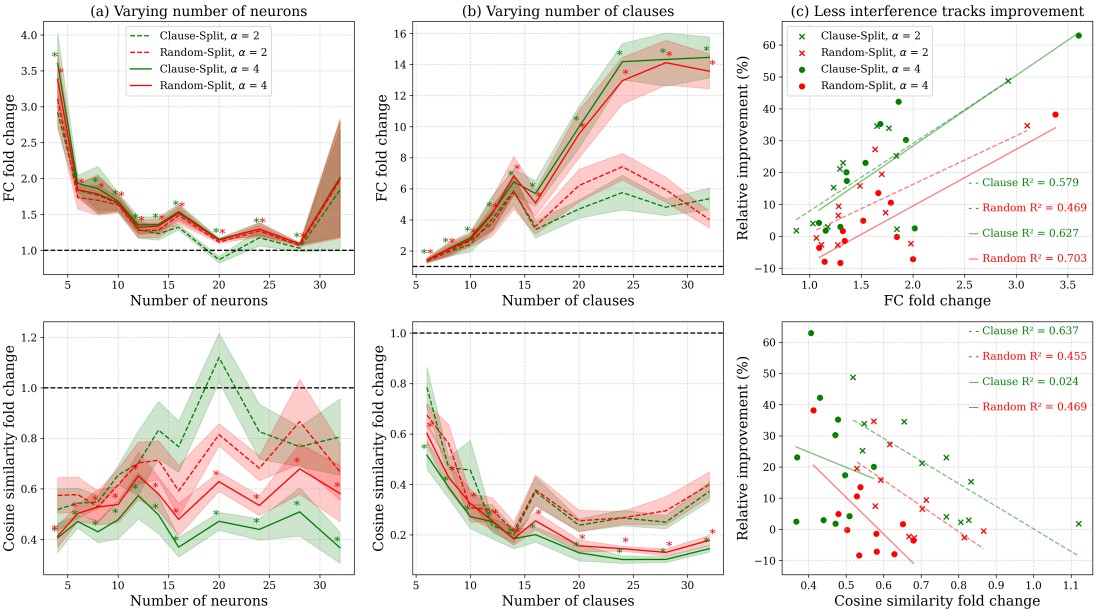

*Figure 3.* Changes in feature interference metrics for varying neurons and clauses. (a) Feature capacity (top) and cosine similarity (bottom) fold change for models of varying hidden dimension on 8 clauses compared to baseline. (b) Feature capacity (top) and cosine similarity (bottom) fold change for models of varying number of clauses for 8 neurons compared to baseline. Legend labels are shared for (a) and (b) and all metrics are normalized to dense values. A fold change of 1.0 represents dense metrics and is shown by a black dashed line. (c) Least-squares regressions on relative improvement versus feature capacity fold change (top) and neuron cosine similarity fold change (bottom) when varying neurons, with the coefficients of determination indicated. Dotted lines correspond to $\alpha = 2$ and solid lines correspond to $\alpha = 4$. Relative improvement is calculated as before. Error bars indicate one standard error of the mean. * indicates $p < 0.05$ and is shown only for $\alpha = 4$ for clarity. Results collected over five trials.

**Feature capacity and cosine similarity**

The top panels of Figure 3a and b demonstrate that FPE consistently increases feature capacity, thus reducing interference, compared to dense baseline models (black dashed line). The benefits when varying neurons are most pronounced in resource-constrained settings, where the dense model struggles with severe superposition. A complementary pattern is observed when varying the number of clauses with 8 neurons fixed. The bottom panels of Figure 3a and b examine neuron cosine similarity, where lower values indicate better feature disentanglement. FPE methods achieve superior orthogonality compared to dense models for all but one setting. Additional experiments in Section A.6.

Furthermore, improvements in performance and interference track strongly while varying width. This suggests that reduced feature interference (more capacity and orthogonality) directly correlates with better accuracy at a fixed parameter count, providing insight into why FPE is beneficial (Figure 3c).

Beyond these global metrics, we also perform explicit circuit and clause level analyses: clause-split FPE models exhibit lower edge-level collision rates and substantially better recovery of the ground-truth clause partition via Gram-matrix clustering than dense baselines, providing more direct evidence of semantic disentanglement (Section A.9).

## 5. Generalization to real-world vision tasks

We evaluate the performance of FPE on four multiclass classification tasks of increasing complexity: FashionMNIST, CIFAR-100, ImageNet-100, and ImageNet-1k.

### 5.1. Fixed representation vision classifiers

We use the single hidden layer classifier (Section 2.1) for the first three tasks, and use a five layer classifier for ImageNet-1k (Section A.8). Inspired by the Gram matrix in the Boolean DNF case, we implement a similar feature-clustering algorithm. As described in Section 2.2, we first train a dense baseline model for 25 epochs to allow for near convergence and compute the feature Gram matrix of $\mathbf{W_1}$. Finally, we cluster the rows of the Gram matrix and treat the clusters as groups of features that should remain together, as is in the case of clauses. We then balance the number of clusters that go to each sub-neuron, and fine-tune for 25 more epochs for both the baseline and the FPE model.

We observe several performance trends for FPE. First, feature-based splitting consistently outperforms the dense baseline across various hidden sizes, affirming the overall effectiveness of the FPE framework (Figure 4). Indeed, in some scenarios, we can double dense performance (Figure 4b), greatly improving accuracy per parameter. How-

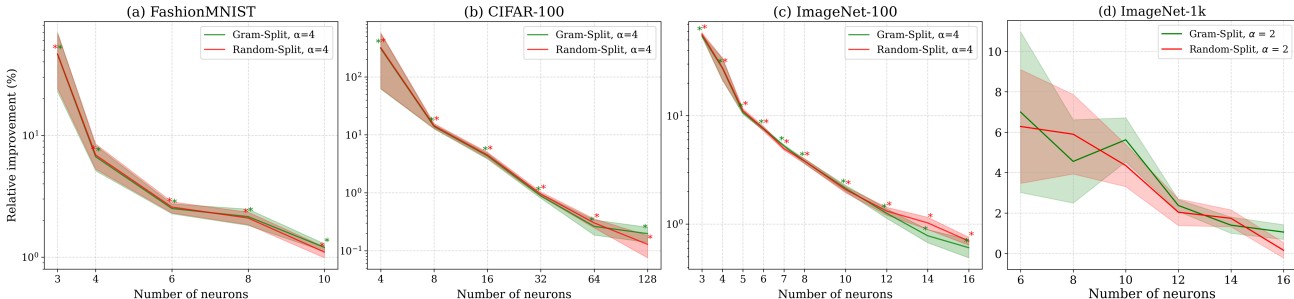

*Figure 4.* Fixed Parameter Expansion helps on real datasets like (a) FashionMNIST, (b) CLIP-embeddings of CIFAR-100, (c) CLIP-embeddings of ImageNet-100, and (d) CLIP-embeddings of ImageNet-1k. In all figures, the number of neurons refers to the number of neurons pre-expansion. Relative improvement is calculated as before. Error bars indicate one standard error of the mean. * indicates $p < 0.05$. Results collected over ten trials.

ever, we find that our feature-based splitting method performs only comparably to random splitting in practice, even though both improve over the dense baseline. This is similar to what we observe in the Boolean DNF experiments (Sections 4.1 and 4.2), where random splitting still provides some gains over the dense baseline.

This consistency across synthetic and real-world settings suggests that FPE confers benefits even when the splitting heuristic does not perfectly align with the underlying structure. It reinforces the idea that increasing representational disentanglement, with or without feature understanding, can mitigate superposition and improve performance, as suggested by our theoretical analysis (Section 3). However, it also suggests that the feature-based splitting algorithm likely did not recover the "true" underlying feature structure of the data. Developing more precise feature attribution or disentanglement methods remains an important avenue for future work. Additionally, the relative improvement of FPE is most pronounced when there are fewer neurons. This too aligns with the hypothesis that FPE is particularly beneficial in settings with high superposition, where neurons are forced to represent multiple overlapping features. As the network widens and has more capacity, the gains from expansion diminish. However, if feature demand scales with network size, FPE gains persist (Figure 6d). Exact test accuracies are provided in Section A.7.

### 5.2. Joint representation and classifier learning

To evaluate whether FPE's interference reducing effects extend beyond frozen representation settings, we also consider when features and the classifier are learned jointly. First, for Boolean DNFs, we prepend a learnable linear embedding layer **E** before the classifier, following Adler et al. (2025). Second, for CIFAR-100, we train a lightweight CNN backbone jointly with a classifier MLP, where the backbone maps each image to an embedding vector that is fed into the MLP. In both cases, the dense model is pre-trained for 25 epochs, then expanded via FPE, and both dense and FPE

variants are fine-tuned for an additional 25 epochs using identical optimization schedules. Unlike the CLIP-based experiments, the feature embeddings here remain fully trainable throughout, allowing us to test whether FPE continues to improve performance when upstream representations are also evolving. More details in Section A.7.1.

In the Boolean setting with a learnable embedding, FPE continues to follow the superposition trends observed in earlier experiments. When we vary the number of neurons for a fixed number of clauses, the relative improvement from FPE is largest at small widths, indicating that its benefits are greatest when superposition pressure is high (Figure 5a). However, when we instead fix the number of neurons and increase the number of clauses, the FPE gains remain much more consistent than in the fixed-feature case: allowing **E** to adapt appears to make the expanded architecture more robust as the number of underlying features grows (Figure 5b). A similar pattern emerges on CIFAR-100 with a trainable CNN backbone and MLP classifier. For both projection sizes, FPE reliably improves accuracy across all tested classifier widths, with the largest relative gains at the smallest widths where the bottleneck is most severe (Figure 5c and d). Together, these results show that FPE's interference reducing advantages persist when representations and classifiers are learned jointly.

### 5.3. Detailed analysis of Fixed Parameter Expansion on CIFAR-100

We conduct additional experiments on CLIP-embeddings of CIFAR-100. Random-split FPE with expansion factor $\alpha = 4$ achieves accuracy comparable to dense models with substantially larger parameter budgets. For example, applying FPE to a dense model of 32 neurons matches one with 1.2 times more non-zero weights, and similar trends hold across widths (Figure 6a). Varying the timing of the split shows that earlier application of FPE yields the best final accuracy, likely by enabling earlier feature specialization, although later splits still outperform dense baselines under the

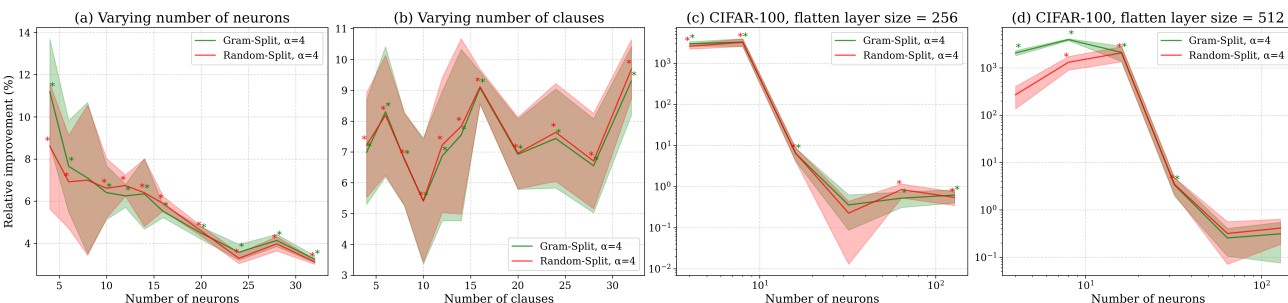

*Figure 5.* Jointly learning and classifying features is improved via increasing width. (a) Training a classifier on a Boolean DNF task with eight clauses of four literals each while varying the number of neurons and with an embedding layer. (b) Training a classifier on a Boolean DNF task with eight neurons while varying the number of clauses, each of which contains four literals, and with an embedding layer. Training a CNN on CIFAR-100 with an embedding dimension of 256 (c) and 512 (d). Relative improvement is calculated as before. Error bars indicate one standard error of the mean. * indicates $p < 0.05$. Results collected over ten trials.

same total training budget (Figure 6b). Fully characterizing the impact of warmup on FPE remains future work. Across model sizes, randomly split FPE models with $\alpha = \{2, 4\}$ consistently exhibit lower mean cosine similarity between neuron weights than dense models, indicating reduced interference that persists as width increases (Figure 6c).

We further probe how FPE gains vary with the balance between feature demand and available neurons in more realistic settings. To isolate this effect, we vary the number of CIFAR-100 classes used for training while also sweeping classifier width. The class subsets are nested, so each larger task strictly contains the smaller ones, yielding a sequence of tasks with monotonically increasing feature demand. Although class count is only a proxy for the number of underlying features, we find that increasing it alongside width maintains substantially stronger FPE gains than in the fixed-class setting, where gains diminish as width increases. At fixed width, larger class counts consistently yield more improvements, matching the prediction that FPE is most useful when superposition pressure is high (Figure 6d). These results complement the Boolean experiments and support the view that FPE targets the feature-to-neuron pressure emphasized by recent superposition analyses (Liu et al., 2025; Adler & Shavit, 2024), rather than simply benefiting smaller models. Exact test accuracies are provided in Section A.7.

Additionally, we validate our framework on deeper architectures and test its compatibility with advanced sparsity techniques such as dynamic mask retraining and structured sparsity, showing that the improvements from FPE apply to such settings as well (Section A.8). We also compare against DropConnect-style (Wan et al., 2013) weight-drop baselines and find that FPE matches or outperforms these methods at equal or larger non-zero parameter budgets (Section A.8.4). These experiments further suggest the potential of improving model performance via interference reduction, especially for larger and more complicated models, which we leave to future work.

## 6. Related Work

Our work connects to several areas of prior research, including superposition, sparse models, and manipulating representational geometry through weight structure. We summarize the connections while highlighting how our goal differs: rather than proposing a new sparsification technique, we study how interference governs model performance.

**Superposition and interpretability**
The superposition hypothesis suggests that networks encode more features than available neurons by representing multiple features within the same neuron. While this allows smaller models to have greater effective capacity, it inherently results in interference, diminishing both interpretability and performance (Olah et al., 2020; Elhage et al., 2022; Liu et al., 2025). Recent formal analyses sharpen this view by showing that modeling a given set of features requires a minimum number of parameters (Adler & Shavit, 2024; Hänni et al., 2024; Adler et al., 2025). Sparse autoencoders (SAEs) take these insights as a starting point and aim to analyze superposition by learning a separate sparse dictionary over activations, often improving semantic interpretability (Brinkmann et al., 2025; Cunningham et al., 2023). FPE is complementary: it uses the same superposition perspective to modify the base network, reallocating a fixed number of weights across more neurons to reduce interference and improve task performance. In principle, SAEs could be applied on top of an FPE network, so we view our method as using motivations from interpretability to guide architecture design rather than competing with SAE-based analyses.

**Network compression and growth**
Previous techniques related to model sparsity either train a large model densely and prune after training (Han et al., 2015; Frantar & Alistarh, 2023; Sun et al., 2023; Fang et al., 2024), or dynamically grow or rewire networks over time (Chen et al., 2015; Yuan et al., 2020; Han et al., 2021; Wu et al., 2020; Pham et al., 2024). We propose a third path

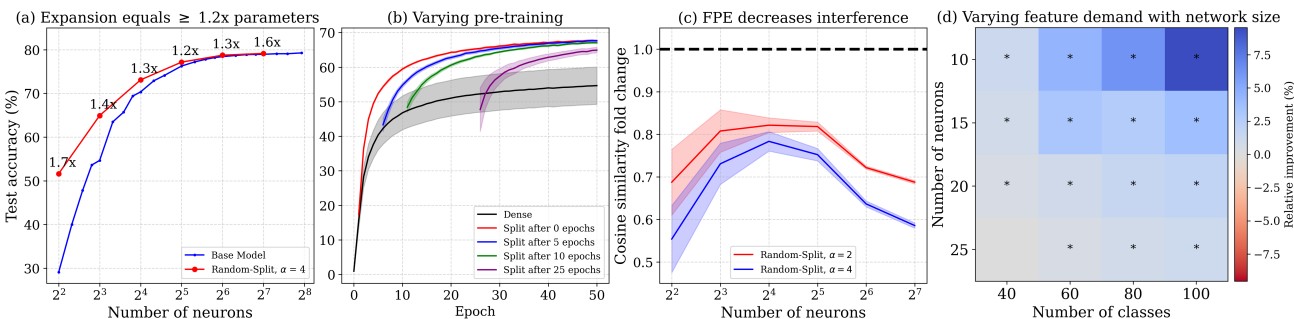

*Figure 6.* More extensive experiments on CIFAR-100 with random splitting. (a) An FPE model with $\alpha = 4$ achieves equivalent performance to dense networks with $\geq 1.2\times$ the number of parameters, which is indicated above each FPE model size. (b) Training curves of dense and random-split models with varying lengths of pre-training for a network with eight original neurons. (c) Randomly-split models for $\alpha = \{2, 4\}$ achieve lower cosine similarity compared to dense baselines (black dashed line) across neuron count, indicating reduced interference. (d) Increasing feature demand with model size maintains FPE improvements more strongly, with $\alpha = 4$ and with * indicating $p < 0.05$. In all figures, the number of neurons refers to the number of neurons pre-expansion. Relative improvement is calculated as before. Error bars indicate one standard error of the mean. Results collected over ten trials.

grounded in insights from superposition: explicitly expanding width at a fixed non-zero parameter budget (Figure A1). Unlike random-masking studies that vary width while training from scratch (Golubeva et al., 2020), FPE is applied after an initial training phase and is designed to reduce interference by assigning features to disjoint sets of weights. DropConnect (Wan et al., 2013) likewise introduces random sparsity, but acts primarily as a regularizer and typically restores dense weights at inference. They do not enforce disjoint sub-neurons or preserve a fixed non-zero budget. By contrast, FPE produces a mechanistically-motivated, deterministic sparse architecture that may also be combined with other compression methods such as quantization (Frantar et al., 2022) and distillation (Boix-Adsera, 2024).

## 7. Conclusion and Limitations

We show that theoretical insights from interpretability can do more than explain trained networks: they can guide architecture design. Across symbolic and real-world settings, increasing width while holding the non-zero parameter budget fixed consistently reduces feature interference and improves accuracy, especially in high-superposition regimes. These regimes arise not only when neuron count is small, but also when feature demand grows alongside model size, so that the ratio of features to neurons remains high. In Boolean tasks, clause-aligned splits reduce polysemanticity and yield the strongest gains, while random splits also improve performance, consistent with our theoretical prediction that simply reducing feature collisions can be beneficial even without precise feature assignments. Results on real models further support this interpretation: the benefits appear to arise primarily from alleviating interference pressure rather than from exactly recovering the underlying feature structure.

Importantly, the gains do not come from adding neurons

alone. FPE preserves the non-zero parameter budget while enforcing disjoint input supports, and a non-disjoint baseline with the same number of sub-neurons and non-zero parameters generally performs worse than FPE (Table A14 and Section A.8.4). Thus, the key intervention is not width expansion in isolation, but reallocating a fixed set of parameters to reduce feature collisions. To our knowledge, this is the first explicit demonstration that reducing superposition is associated with improved performance at a fixed non-zero parameter count.

At the same time, FPE is best viewed as a mechanistic proof of concept rather than a deployment ready method, and several limitations remain. First, although fixed non-zero parameter count is attractive on modern accelerators, where memory movement of non-zero parameters can be a primary bottleneck (Rajbhandari et al., 2021), practical speedups will depend on sparse kernel support, activation overhead, and hardware specific implementation. Second, our feature-based splits on real data rely on approximate heuristics rather than true feature decompositions. Stronger feature discovery methods, such as SAEs, may produce more precise partitions and larger gains. Third, FPE is most useful when interference is high, either because neuron count is limited or feature demand is large, and its benefits naturally diminish when the dense model already has sufficient capacity. Finally, our experiments focus on controlled and relatively small-scale settings, so extending FPE to depth-wise expansions and larger, transformer-scale models remains an important direction for future work.

Overall, our results identify neuron count as an architectural axis distinct from parameter count. By redistributing a fixed non-zero parameter budget across more specialized units, FPE provides an interpretability-grounded mechanism for improving performance through reduced superposition.

## Acknowledgments

The authors would like to thank Dan Gutfreund, Alexandre Marques, Shashata Sawmya, Tony Wang, Thomas Athey, Timothy Gomez, and Benjamin Cohen-Wang for providing valuable feedback and constructive guidance as the project developed.

This project was supported by an MIT-IBM Watson AI Lab grant, an NIH Brains CONNECTS U01 grant, and the MIT UROP Office.

## Impact Statement

This paper presents work whose goal is to advance the field of Machine Learning. We specifically seek to better understand how to leverage superposition for increased performance for a given number of parameters. The potential societal consequences of our work include more sustainable and efficient machine learning systems.

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

# A. Appendix

## A.1. Fixed Parameter Expansion framework

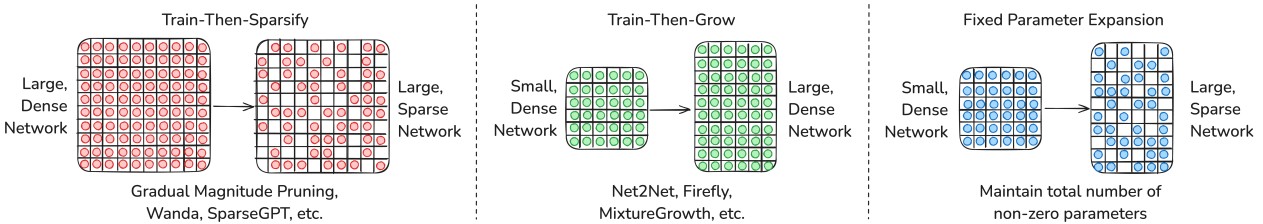

*Figure A1.* Paradigms of parameter efficiency in training and inference. "Train-then-sparsify" minimizes post-training inference cost, at the expense of training a large, dense model initially and of sparse fine-tuning. "Train-then-grow" amortizes some training cost via bootstrapping from a small, dense network, to yield a large, dense network that is more expensive to inference, though some techniques are constrained by final size. Here, we show theoretical justification and empirical feasibility of directly splitting neurons within a small, dense network into a large, sparse one.

## A.2. Fixed Parameter Expansion pseudocode

---

**Algorithm A1** Fixed Parameter Expansion via Neuron Splitting

---

**Require:** Weight matrices $\mathbf{W_1} \in \mathbb{R}^{h \times d}$, $\mathbf{W_2} \in \mathbb{R}^{C \times h}$, expansion factor $\alpha \in \mathbb{N}$
**Ensure:** Expanded, sparse matrices $\tilde{\mathbf{W}}_1, \tilde{\mathbf{W}}_2$ with the same total number of non-zero parameters
 1: Let $h' \leftarrow \alpha h$; initialize $\tilde{\mathbf{W}}_1 \in \mathbb{R}^{h' \times d}$ and sparsity mask $\mathbf{M_1} \in \{0,1\}^{h' \times d}$
 2: **for** $i = 1$ to $h$ **do**
 3:     Duplicate neuron $i$ into $\alpha$ sub-neurons $i_1, \ldots, i_\alpha$
 4:     Partition $\mathbf{1}_d$ into disjoint masks $m^{(i_j)} \in \{0,1\}^d$ such that: $\sum_{j=1}^{\alpha} m^{(i_j)} = \mathbf{1}_d$
 5:     Assign each $m^{(i_j)} \cdot \mathbf{W_1}[i,:]$ to row $i * h + j$ in $\tilde{\mathbf{W}}_1$ and each $m^{(i_j)}$ to row $i * h + j$ in $\mathbf{M_1}$
 6: **end for**
 7: Initialize $\tilde{\mathbf{W}}_2 \in \mathbb{R}^{C \times h'}$ and sparsity masks $\mathbf{M'_1} \in \{0,1\}^{h' \times d}$ and $\mathbf{M'_2} \in \{0,1\}^{C \times h'}$
 8: Estimate $\Delta \leftarrow (\alpha - 1)h \cdot C$
 9: Prune smallest-magnitude weights in $\tilde{\mathbf{W}}_1$ and $\tilde{\mathbf{W}}_2$ to enforce total parameter count $\leq P$
10: **return** $\tilde{\mathbf{W}}_1, \tilde{\mathbf{W}}_2$

---

## A.3. Training and dataset details

### A.3.1. BOOLEAN DNF DATASET GENERATOR

For our experiments on Boolean satisfiability, we construct a binary classification dataset over $m$ Boolean variables partitioned into $C = \frac{m}{k}$ disjoint clauses of size $k$. Positive examples (half of the data) are guaranteed to satisfy at least one clause (i.e. all $k$ literals in that clause are set to 1), by turning on all $k$ literals in a randomly chosen clause, then adding extra "1"s until the total number of active bits lies between $\lfloor \frac{m}{4} \rfloor$ and $\lfloor \frac{m}{4} \rfloor + \lfloor \frac{m}{8} \rfloor$. Negative examples are crafted to violate every clause: we either start from a candidate satisfying assignment and flip one literal per clause, or sample arbitrary assignments and reject any that accidentally satisfy a clause. We retry until we have exactly half positive and half negative samples, with equal proportions of the negative samples generated by flipping the literal and randomly sampling. A fixed random seed ensures full reproducibility. Finally, to improve training stability and encourage generalization, we introduce small continuous variability into the binary input vectors: bits set to `True` (1) are mapped to random values in the range $[3, 3.5]$, while `False` (0) bits are mapped to values in $[0, 0.5]$. This preserves the logical structure of the satisfiability task while ensuring sufficient activation magnitude for effective learning in ReLU networks. Output labels remain binary.

---

**Algorithm A2** Generate Boolean DNF Classification Data

---

**Require:** $n$ total samples, $m$ total literals, $k$ literals per clause, positive data active bits range (minOnes $= \lfloor m/4 \rfloor$, maxOnes $= \lfloor m/4 \rfloor + \lfloor m/8 \rfloor$), positive data proportion $p = 0.5$, seed (optional)
**Ensure:** $\mathbf{X} \in \{0, 1\}^{n \times m}, \mathbf{y} \in \{0, 1\}^n$
 1: **if** seed $\neq \varnothing$ **then** set all RNGs to seed
 2: $n_+ \leftarrow \lfloor p\,n \rfloor, \quad n_- \leftarrow n - n_+$
 3: $C \leftarrow m\,/\,k$
 4: Initialize $\mathbf{X}_+ \in \{0\}^{n_+ \times m}, \mathbf{X}_- \in \{0\}^{n_- \times m}$
 5: Initialize $\mathbf{y}_+ \leftarrow \mathbf{1}_{n_+}, \mathbf{y}_- \leftarrow \mathbf{0}_{n_-}$
 6: **for** $i = 1$ to $n_+$ **do**
 7:     Draw clause index $c \sim \text{Uniform}\{0, \ldots, C - 1\}$
 8:     Draw $s \sim \text{UniformInt}(\text{minOnes}, \text{maxOnes})$
 9:     $L \leftarrow \{c \cdot k, \ldots, c \cdot k + k - 1\}$
10:     Add $(s - k)$ random indices from $\{0, \ldots, m - 1\} \setminus L$ into $L$
11:     Set $\mathbf{X}_+[i, L] \leftarrow 1$
12: **end for**
13: $i_- \leftarrow 1, u \leftarrow -1$
14: **while** $i_- \leq n_-$ **do**
15:     Flip a fair coin:
16:     **if** heads **then**
17:         Pick clause $c$ and provisional sparsity $s \leftarrow (u \geq 0?s : u'), u' \sim \text{UniformInt}(\text{minOnes}, \text{maxOnes})$
18:         Build $L$ as above, then remove one random index from the clause-block
19:     **else**
20:         $s \leftarrow (u \geq 0?s : u')$, sample $L$ uniformly of size $s$ from all $m$ literals
21:     **end if**
22:     Set $\mathbf{X}_-[i_-, L] \leftarrow 1$
23:     **if** EVALUATEDNF$(\mathbf{X}_-[i_-], m, k) = 0$ **then**
24:         $i_- \leftarrow i_- + 1, u \leftarrow -1$
25:     **else**
26:         Reset $\mathbf{X}_-[i_-] \leftarrow \mathbf{0}, u \leftarrow s$
27:     **end if**
28: **end while**
29: **return** $[\mathbf{X}_+; \mathbf{X}_-], [\mathbf{y}_+; \mathbf{y}_-]$

---

### A.3.2. VISUAL TASKS

1. **FashionMNIST** consists of 10 clothing classes with balanced class distributions. We use the standard train/test split.

2. **CIFAR-100** includes 100 object categories with 600 images each. We use both the raw as well as the CLIP-extracted representations of the training and test sets for classification.

3. **ImageNet-100** is a 100-class subset of ImageNet-1k, which we selected randomly for computational tractability, as explained in more detail in Section 2.2. We extract CLIP embeddings for both the training and validation splits.

4. **ImageNet-1k** is a widely-used subset of the larger ImageNet dataset, containing approximately 1.2 million training images from 1,000 distinct object classes. We use CLIP-extracted embeddings for both training and validation splits.

### A.3.3. TRAINING AND PARAMETERS

In our experiments, we compare three types of models: base neural network, Gram-split model, and a random-split model (Section 5). Training was performed on 6 RTX 2080 Ti GPU's, although these tasks are not demanding and could be run on CPU. The number of pre-training epochs was chosen so that at the time of the FPE split, the model had reached performance at least within $95\%$ of that of a fully converged model across all datasets.

All models are trained on the Boolean DNF data (Algorithm A2), with identical hyperparameters:

- Optimizer: Adam with learning rate $\eta = 10^{-3}$

- Batch size B = 64

- Number of pre-training epochs: 25

- Number of fine-tuning epochs: 25

- Regularization: $\lambda_1 = 10^{-7}$ on $\mathbf{W_1}$, $\lambda_2 = 10^{-5}$ on all parameters.

- Trials T = 10

The only exception to this training procedure is in Figure 1, where 1000 epochs of pre-training and fine-tuning were used to fully develop the clauses for the purposes of clearer visualization, and in Figures 2 and 3, where five trials were used. All other experiments, both those focused on interpretability and on performance, were conducted with the hyperparameters above.

---

**Algorithm A3** Model Training and Selection

---

**Require:** Training data $(X_{\mathrm{tr}}, y_{\mathrm{tr}})$, Test data $(X_{\mathrm{te}}, y_{\mathrm{te}})$

1: Model class $\mathcal{M} \in \{\mathrm{BaseModel}, \mathrm{GramSplitModel}, \mathrm{RandomSplitModel}\}$ pre-training epochs $P$, fine-tuning epochs $F$, batch size $B$, learning rate $\eta$, regularization penalties $\lambda_1, \lambda_2, \lambda_{\mathrm{orth}}$, trials $T$

**Ensure:** Best model $\hat{M}$ and its test accuracy

2:   bestAcc $\leftarrow 0$
3: **for** $t = 1, \ldots, T$ **do**
4:     Instantiate $M \leftarrow$ BaseModel and move to device
5:     opt $\leftarrow \mathrm{Adam}(M.\mathrm{parameters}(), \mathrm{lr} = \eta)$
6:     Define binary cross-entropy loss BCE
7:     Build a DataLoader over $(X_{\mathrm{tr}}, y_{\mathrm{tr}})$ with batch size $B$
8:     **for** epoch $e = 1$ to $P$ **do**
9:       **for** each batch $(x_b, y_b)$ **do**
10:         opt.zero_grad()
11:         $\hat{y} \leftarrow M(x_b)$
12:         $\ell \leftarrow \mathrm{BCE}(\hat{y}, y_b)$
13:         **if** $\lambda_1 > 0$ **then**
14:           $\ell + = \lambda_1 \|W_1\|_1$
15:         **end if**
16:         **if** $\lambda_2 > 0$ **then**
17:           $\ell + = \lambda_2 \sum_{p \in M.\mathrm{parameters}} \|p\|_2^2$
18:         **end if**
19:         **if** $\lambda_{\mathrm{orth}} > 0$ **then**
20:           $\ell + = \lambda_{\mathrm{orth}} \|W_1 W_1^T - I\|_F^2$
21:         **end if**
22:         $\ell$.backward() opt.step()
23:       **end for**
24:     **end for**
25:     Either keep fine-tuning the BaseModel or split the model via FPE
26:     **for** epoch $e = P + 1$ to $P + F$ **do**
27:       **for** each batch $(x_b, y_b)$ **do**
28:         opt.zero_grad()
29:         $\hat{y} \leftarrow M(x_b)$
30:         $\ell \leftarrow \mathrm{BCE}(\hat{y}, y_b)$
31:         **if** $\lambda_1 > 0$ **then**
32:           $\ell + = \lambda_1 \|W_1\|_1$
33:         **end if**
34:         **if** $\lambda_2 > 0$ **then**
35:           $\ell + = \lambda_2 \sum_{p \in M.\mathrm{parameters}} \|p\|_2^2$
36:         **end if**
37:         **if** $\lambda_{\mathrm{orth}} > 0$ **then**
38:           $\ell + = \lambda_{\mathrm{orth}} \|W_1 W_1^T - I\|_F^2$
39:         **end if**
40:         $\ell$.backward() opt.step()
41:       **end for**
42:     **end for**
43:     {At each epoch, evaluate on test set}
44:     $\hat{y}_{\mathrm{te}} \leftarrow M(X_{\mathrm{te}})$
45:     acc $\leftarrow \mathrm{mean}\big(\arg\max_k \hat{y}_{\mathrm{te},k} = y_{\mathrm{te}}\big)$
46:     **if** acc $>$ bestAcc **then**
47:       bestAcc $\leftarrow$ acc,    $\hat{M} \leftarrow M$
48:     **end if**
49: **end for**
50: **return** $\hat{M}$, bestAcc

---

## A.4. Even Random-Split FPE reduces interference and improves performance

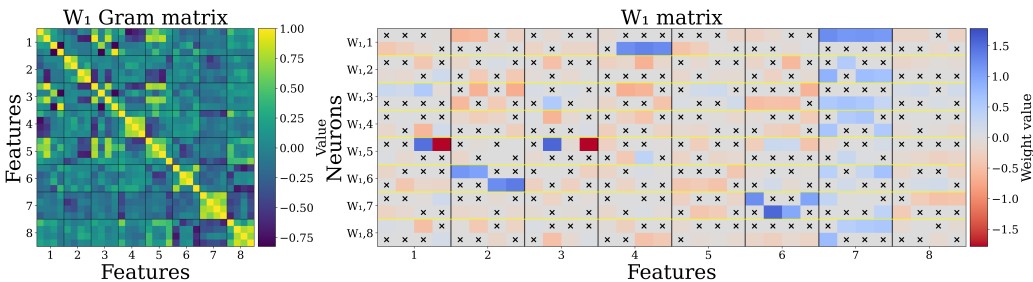

*Figure A2.* Random-split model gram and $\mathbf{W_1}$ matrices. Codes are more broken between neurons. Black x's represent masked parameters of value 0.

### A.5. Theoretical justification for Fixed Parameter Expansion

We demonstrate theoretical properties of a randomly constructed Fixed Parameter Expansion (FPE) network. We find that with high probability, an FPE network maintains "coverage" of all clauses by at least one neuron, while significantly reducing the interference caused by a single neuron covering multiple clauses, when compared to a dense network with the same number of non-zero parameters. This holds true even when the FPE network is constructed by randomly choosing a subset of active weights for each neuron without any knowledge of the task structure.

**Model Setup**    We analyze a DNF formula with $C$ clauses over $m$ literals, with each clause having $k$ literals. We compare two architectures with equal parameter counts. One is a dense network with $r$ neurons, each connected to all $m$ literals. The other is a Fixed Parameter Expansion (FPE) network with expansion factor $\alpha$, with $\alpha r$ neurons, each of which is connected to $d = \frac{m}{\alpha}$ literals by non-zero parameters, chosen at random for each neuron. Crucially, this selection is performed without any knowledge of the underlying clause structure. For simplicity, let us allow for replacement when choosing non-zero weights between all neurons. Thus, the argument does not require the disjoint-support constraint used by FPE, and is closer to the relaxed non-disjoint baseline in Section A.8.4. We say a neuron covers a clause if the neuron's weights for each literal in the clause has the potential to be non-zero.

**Clause Coverage Probability**    First, for the network to learn the DNF formula, there must be at least one neuron that can compute each clause. A neuron can only compute a clause if it receives input from all the literals that comprise it; otherwise, the clause is fundamentally incomputable for that neuron. Therefore, we first show that the FPE network is highly likely to cover each clause, even though the network's sparse connections were chosen randomly.

The probability $p$ that a single sparse neuron covers a specific $k$-literal clause is $p = \frac{\binom{m-k}{d-k}}{\binom{m}{d}}$. For a single clause $S_i$, the probability that it is not covered by one neuron is $(1-p)$. Because of our independence assumption, the probability that $S_i$ is missed by every neuron is Pr[clause $S_i$ missed] $M_i = (1-p)^{\alpha r}$. To find the probability that at least one of the $C$ clauses is missed, we can use a union bound to provide an upper bound on the probability that at least one of the C clauses is missed. Pr[at least one clause missed] $= \Pr(\bigcup_{i=1}^{C} M_i) \leq \sum_{i=1}^{C} \Pr[M_i] = C \cdot (1-p)^{\alpha r}$. The probability that all clauses are covered Pr[all clauses covered] is therefore $\geq 1 - C \cdot (1-p)^{\alpha r}$. For large $m$ and $d$, $p$ is well approximated by $p = \frac{\binom{m-k}{d-k}}{\binom{m}{d}} = \frac{d!(m-k)!}{m!(d-k)!} \approx (\frac{d}{m})^k = (\frac{1}{\alpha})^k$, and for small $x$, $(1-x) \approx e^{-x}$. Together, Pr[all clauses covered] $\geq 1 - C \cdot (1-p)^{\alpha r} \approx 1 - C \cdot e^{-r/\alpha^{k-1}}$. To ensure Pr[at least one clause missed] $\leq C \cdot e^{-r/\alpha^{k-1}}$ is within some tolerance $\epsilon$, $C \cdot e^{-r/\alpha^{k-1}} < \epsilon$. Solving for $r$ yields $r > \alpha^{k-1} \ln(\frac{C}{\epsilon})$. This provides a concrete requirement on the network's dimensions to guarantee clause coverage with high probability.

**Interference Analysis**    Now, let us consider the decrease in interference from this procedure. We analyze interference because it is a proxy for the difficulty of the optimization problem. When a single neuron covers multiple clauses, its weights receive conflicting gradient updates during training, as it tries to simultaneously represent multiple distinct concepts. This entanglement can make it difficult for optimizers to learn a clean representation for any single clause. By showing that FPE dramatically reduces the expected number of these interference events, we argue that it creates a more factored, disentangled learning problem that is fundamentally easier to solve.

For two clauses $S$ and $T$, in the dense network, the probability that neuron $r$ covers both is 1. Thus, the total number of interference instances (a neuron covering a pair of clauses) is $E_{dense} = r \cdot \binom{C}{2}$. On the other hand, in the FPE network, the probability $p'$ that a neuron $r$ covers both $S$ and $T$ is $p' = \frac{\binom{m-2k}{d-2k}}{\binom{m}{d}} \approx (\frac{1}{\alpha})^{2k}$. The expected number of clause collisions in an FPE network is therefore $E_{FPE} = \alpha r \cdot \binom{C}{2} \cdot p'$. The ratio of the expected number of collisions in the FPE network to that of dense network is therefore $\frac{E_{FPE}}{E_{dense}} = \frac{\alpha r \cdot \binom{C}{2} \cdot p'}{r \cdot \binom{C}{2}} = \alpha p' \approx \frac{1}{\alpha^{2k-1}}$. This ratio of interference shows a significant reduction for the FPE network. Intuitively, for a total of $2k$ literals to be covered by the same neuron, the first literal is free to be placed onto any neuron, while the subsequent $2k - 1$ literals must be in the $\frac{1}{\alpha}$ non-zero weights for the neuron. The approximate probability of this happening is $\frac{1}{\alpha^{2k-1}}$. Thus, our analysis shows that a sparse, wide FPE network is theoretically advantageous, even in the case where the clauses are not explicitly known. By simply applying random sparsity, it maintains coverage of all logical clauses while drastically reducing interference, all without needing to know the specific clauses in advance.

### A.6. Additional feature interference reduction measurements

*Table A1*. Average total feature capacity. Higher values represent decreased superposition. Each network originally had eight dense neurons. Results averaged across five trials.

| # Literals | Dense | Clause-split | Random-split |
|---|---|---|---|
| 12 | $1.896 \pm 0.377$ | $2.740 \pm 0.114$ | $2.780 \pm 0.133$ |
| 24 | $2.907 \pm 0.370$ | $5.479 \pm 0.400$ | $5.268 \pm 0.317$ |
| 32 | $3.944 \pm 0.409$ | $6.977 \pm 0.426$ | $6.848 \pm 0.568$ |
| 40 | $4.819 \pm 0.491$ | $7.853 \pm 0.469$ | $7.825 \pm 0.365$ |
| 60 | $5.584 \pm 0.670$ | $10.79 \pm 0.868$ | $10.90 \pm 0.747$ |
| 80 | $5.812 \pm 1.044$ | $13.07 \pm 1.093$ | $12.00 \pm 1.726$ |
| 100 | $6.219 \pm 0.535$ | $14.21 \pm 1.866$ | $15.00 \pm 0.727$ |
| 128 | $7.699 \pm 1.029$ | $16.87 \pm 0.926$ | $13.80 \pm 1.037$ |

*Table A2*. Average neuron cosine similarity. Lower values represent decreased superposition. Each network originally had eight dense neurons. Results averaged across five trials.

| # Literals | Dense | Clause-split | Random-split |
|---|---|---|---|
| 12 | $0.332 \pm 0.123$ | $0.230 \pm 0.021$ | $0.243 \pm 0.083$ |
| 24 | $0.382 \pm 0.127$ | $0.173 \pm 0.041$ | $0.219 \pm 0.067$ |
| 32 | $0.291 \pm 0.230$ | $0.150 \pm 0.072$ | $0.171 \pm 0.092$ |
| 40 | $0.257 \pm 0.096$ | $0.137 \pm 0.019$ | $0.146 \pm 0.049$ |
| 60 | $0.275 \pm 0.095$ | $0.145 \pm 0.030$ | $0.142 \pm 0.045$ |
| 80 | $0.303 \pm 0.101$ | $0.108 \pm 0.035$ | $0.113 \pm 0.025$ |
| 100 | $0.312 \pm 0.081$ | $0.134 \pm 0.028$ | $0.144 \pm 0.041$ |
| 128 | $0.262 \pm 0.081$ | $0.105 \pm 0.031$ | $0.108 \pm 0.027$ |

## A.7. Raw test accuracies for real-world vision tasks

Here, we show the accuracy per parameter for models presented in Figures 2, 4 and 5. Additionally, we provide the raw test accuracies associated with Figures 4 and 5.

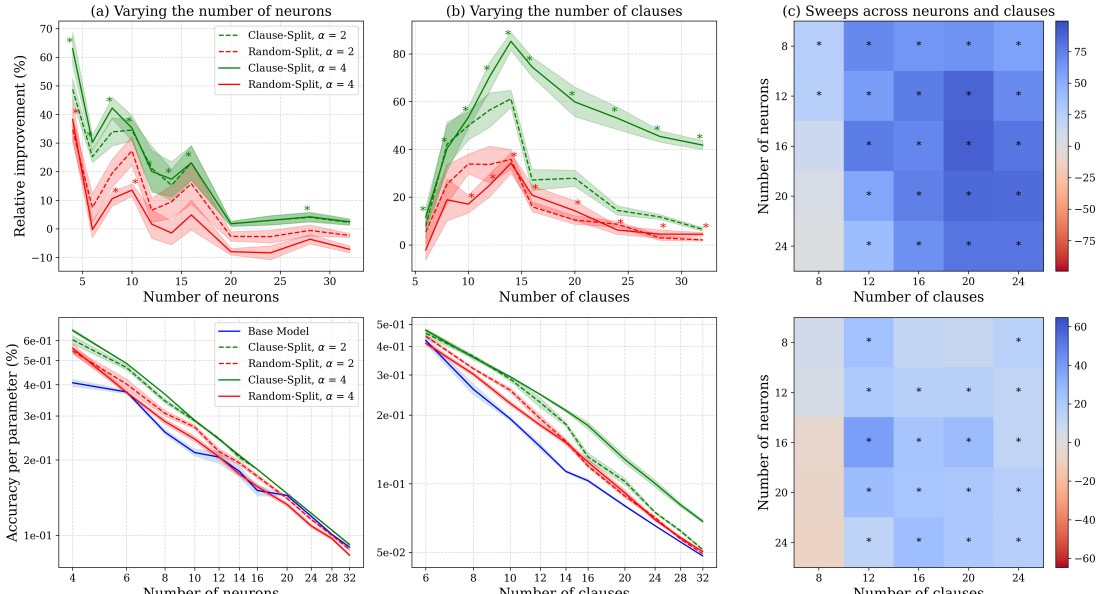

*Figure A3.* Trends of performance under superposition in neurons and clauses. (a) Relative improvement in test accuracy and the accuracy per parameter for models of varying hidden dimension on 8 clauses. (b) Relative improvement in test accuracy and the accuracy per parameter for models with 8 neurons. y-axis labels are shared for (a) and (b). (c) Heatmap of relative improvement percentage for clause-split FPE models (top) and random-split FPE models (bottom). Relative improvement is calculated as $\frac{\text{FPE test accuracy} - \text{dense test accuracy}}{\text{dense test accuracy}}$. Error bars indicate one standard error of the mean. * indicates $p < 0.05$ and is shown only for $\alpha = 4$ for clarity. Results collected over five trials.

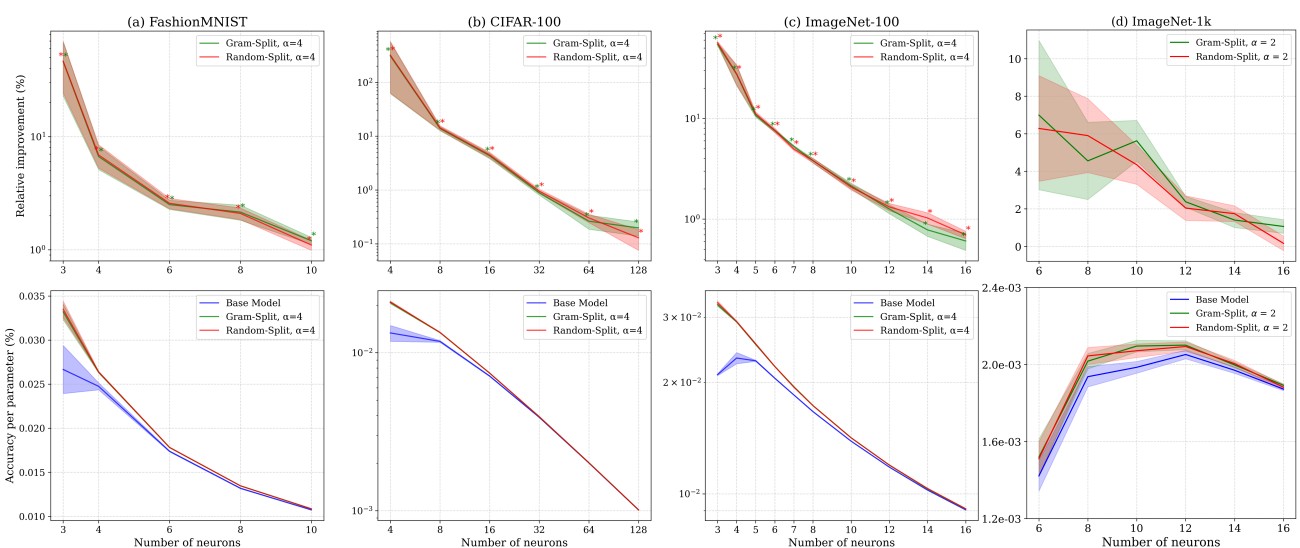

*Figure A4.* Fixed Parameter Expansion helps on real datasets like (a) FashionMNIST, (b) CLIP-embeddings of CIFAR-100, (c) CLIP-embeddings of ImageNet-100, and (d) CLIP-embeddings of ImageNet-1k. All baseline models were trained for 25 epochs before FPE. After the split, both the FPE model and dense model were further fine-tuned for an additional 25 epochs. The first row indicates the improvement in test accuracy of the FPE model relative to the dense baseline, for varying hidden dimensions, and is calculated as before. The second row shows the test accuracy per parameter for each configuration. In all figures, the number of neurons refers to the number of neurons pre-expansion. Relative improvement is calculated as before. Error bars indicate one standard error of the mean. * indicates $p < 0.05$. Results collected over ten trials.

*Table A3.* FashionMNIST. Results averaged across 10 trials.

| Hidden size | Dense | Gram-Split FPE | Random-Split FPE |
|:---:|:---:|:---:|:---:|
| 3 | $63.5 \pm 6.5$ | $79.1 \pm 2.1$ | $\mathbf{79.8 \pm 2.2}$ |
| 4 | $78.6 \pm 1.2$ | $83.7 \pm 0.2$ | $\mathbf{83.8 \pm 0.2}$ |
| 6 | $82.8 \pm 0.3$ | $\mathbf{84.9 \pm 0.1}$ | $\mathbf{84.9 \pm 0.1}$ |
| 8 | $83.8 \pm 0.3$ | $\mathbf{85.5 \pm 0.1}$ | $\mathbf{85.5 \pm 0.1}$ |
| 10 | $85.2 \pm 0.1$ | $\mathbf{86.2 \pm 0.1}$ | $86.1 \pm 0.1$ |

*Table A4.* CLIP-embeddings of CIFAR-100. Results averaged across 10 trials.

| Hidden size | Dense | Gram-Split FPE | Random-Split FPE |
|:---:|:---:|:---:|:---:|
| 4 | $32.7 \pm 3.8$ | $50.8 \pm 0.9$ | $\mathbf{51.3 \pm 0.7}$ |
| 8 | $57.8 \pm 0.7$ | $65.7 \pm 0.2$ | $\mathbf{65.9 \pm 0.1}$ |
| 16 | $69.7 \pm 0.5$ | $72.7 \pm 0.2$ | $\mathbf{72.8 \pm 0.1}$ |
| 32 | $76.4 \pm 0.1$ | $77.0 \pm 0.1$ | $\mathbf{77.1 \pm 0.1}$ |
| 64 | $78.6 \pm 0.1$ | $\mathbf{78.8 \pm 0.0}$ | $\mathbf{78.8 \pm 0.1}$ |
| 128 | $79.2 \pm 0.0$ | $\mathbf{79.4 \pm 0.0}$ | $79.3 \pm 0.0$ |

*Table A5.* CLIP-embeddings of ImageNet-100. Results averaged across 10 trials.

| Hidden size | Dense | Gram-Split FPE | Random-Split FPE |
|---|---|---|---|
| 3 | $38.5 \pm 0.2$ | $59.5 \pm 0.5$ | $\mathbf{60.3 \pm 0.6}$ |
| 4 | $57.0 \pm 2.0$ | $71.4 \pm 0.3$ | $\mathbf{71.5 \pm 0.3}$ |
| 5 | $70.0 \pm 0.2$ | $77.5 \pm 0.2$ | $\mathbf{77.8 \pm 0.2}$ |
| 6 | $75.4 \pm 0.2$ | $\mathbf{81.1 \pm 0.1}$ | $\mathbf{81.1 \pm 0.2}$ |
| 7 | $79.2 \pm 0.1$ | $\mathbf{83.4 \pm 0.1}$ | $83.2 \pm 0.1$ |
| 8 | $81.6 \pm 0.2$ | $\mathbf{84.7 \pm 0.1}$ | $\mathbf{84.7 \pm 0.1}$ |
| 10 | $84.9 \pm 0.1$ | $\mathbf{86.7 \pm 0.1}$ | $86.6 \pm 0.1$ |
| 12 | $86.7 \pm 0.1$ | $\mathbf{87.8 \pm 0.1}$ | $\mathbf{87.8 \pm 0.1}$ |
| 14 | $87.7 \pm 0.1$ | $88.4 \pm 0.1$ | $\mathbf{88.6 \pm 0.1}$ |
| 16 | $88.6 \pm 0.1$ | $89.1 \pm 0.1$ | $\mathbf{89.2 \pm 0.0}$ |

*Table A6.* CLIP-embeddings of ImageNet-1k using a 5-layer MLP. Results averaged across 10 trials.

| Hidden size | Dense | Random-Split FPE |
|---|---|---|
| 6 | 13.0 | **13.9** |
| 8 | 23.8 | **25.1** |
| 10 | 30.6 | **31.9** |
| 12 | 38.1 | **38.8** |
| 14 | 42.9 | **43.6** |
| 16 | 46.9 | **47.0** |

*Table A7.* CLIP-embeddings of CIFAR-100, varying both the number of neurons and the number of classes. Results averaged across 10 trials.

Dense

| Hidden size | 40 | 60 | 80 | 100 |
|---|---|---|---|---|
| 10 | $76.92 \pm 0.32$ | $71.43 \pm 0.51$ | $68.10 \pm 0.94$ | $62.87 \pm 1.06$ |
| 15 | $79.82 \pm 0.30$ | $75.55 \pm 0.87$ | $74.06 \pm 0.47$ | $70.11 \pm 0.31$ |
| 20 | $81.44 \pm 0.09$ | $78.99 \pm 0.22$ | $76.32 \pm 0.26$ | $73.48 \pm 0.19$ |
| 25 | $82.36 \pm 0.18$ | $80.21 \pm 0.10$ | $78.29 \pm 0.15$ | $75.26 \pm 0.20$ |

Gram-Split FPE

| Hidden size | 40 | 60 | 80 | 100 |
|---|---|---|---|---|
| 10 | $\mathbf{77.81 \pm 0.16}$ | $74.53 \pm 0.19$ | $72.20 \pm 0.18$ | $68.68 \pm 0.16$ |
| 15 | $\mathbf{80.47 \pm 0.14}$ | $\mathbf{77.87 \pm 0.21}$ | $\mathbf{76.00 \pm 0.24}$ | $\mathbf{72.69 \pm 0.13}$ |
| 20 | $\mathbf{81.84 \pm 0.08}$ | $\mathbf{79.75 \pm 0.13}$ | $77.50 \pm 0.08$ | $74.80 \pm 0.12$ |
| 25 | $82.38 \pm 0.11$ | $80.58 \pm 0.09$ | $\mathbf{78.98 \pm 0.09}$ | $76.05 \pm 0.09$ |

Random-Split FPE

| Hidden size | 40 | 60 | 80 | 100 |
|---|---|---|---|---|
| 10 | $77.74 \pm 0.15$ | $\mathbf{74.72 \pm 0.20}$ | $\mathbf{72.41 \pm 0.15}$ | $\mathbf{68.77 \pm 0.20}$ |
| 15 | $\mathbf{80.47 \pm 0.14}$ | $77.86 \pm 0.24$ | $75.89 \pm 0.18$ | $72.64 \pm 0.14$ |
| 20 | $81.67 \pm 0.09$ | $79.64 \pm 0.12$ | $\mathbf{77.59 \pm 0.14}$ | $\mathbf{74.95 \pm 0.12}$ |
| 25 | $\mathbf{82.51 \pm 0.07}$ | $\mathbf{80.72 \pm 0.09}$ | $78.91 \pm 0.08$ | $\mathbf{76.19 \pm 0.11}$ |

A.7.1. LEARNING FEATURES JOINTLY WITH CLASSIFICATION

Recall that for our simple classifier, we use $\mathbf{z} = \mathbf{W_2} \cdot \text{ReLU}(\mathbf{W_1}\mathbf{x} + \mathbf{b}_1) + \mathbf{b}_2$, where $\mathbf{x} \in \mathbb{R}^d$ is the input and $h$ denotes the number of neurons. To investigate how FPE performs when jointly learning features and classification, we experiment with two avenues. In both cases, the entire model is pre-trained densely for 25 epochs, split via FPE, and fine-tuned for another 25 epochs, and compared to a model that was trained densely for 50 epochs. The feature embeddings are not frozen at any phase.

**Boolean Experiments with Embedding Layer**  First, for our Boolean experiments, we can prepend a linear embedding layer $\mathbf{E}$ before the classifier to act as a feature learner. Now, $\mathbf{z} = \mathbf{W_2} \cdot \text{ReLU}(\mathbf{W_1}\mathbf{E}\mathbf{x} + \mathbf{b}_1) + \mathbf{b}_2$. We can expand $\mathbf{W_1}$ and $\mathbf{W_2}$ as before. When either varying the number of neurons for eight clauses (Figure 5a), or varying the number of clauses for eight neurons (Figure 5b), FPE delivers consistent improvement in performance to the dense model.

**CIFAR-100 Images with CNN Backbone**  Second, we use a CNN backbone with an MLP block. The network operates on 3-channel images of spatial resolution 32×32, and we use it to classify CIFAR-100. The CNN backbone consists of three VGG-style (Simonyan & Zisserman, 2014) blocks with increasing channel width, followed by a 1 by 1 projection and global average pooling. This is prepended to an MLP block, that can be split via FPE as before. In more detail, the three blocks are as follows:

1. **Convolutional blocks.** Each block maps an input with $C_{\text{in}}$ channels to an output with $C_{\text{out}}$ channels using a stack of three $3 \times 3$ convolutions with same padding, each followed by batch normalization and a ReLU nonlinearity:

$$\text{Conv2d}(C_{\text{in}}, C_{\text{out}}, \text{kernel} = 3, \text{stride} = 1, \text{padding} = 1, \text{bias} = \text{False})$$
$$\rightarrow \text{BatchNorm2d}(C_{\text{out}}) \rightarrow \text{ReLU},$$
$$\text{Conv2d}(C_{\text{out}}, C_{\text{out}}, \text{kernel} = 3, \text{stride} = 1, \text{padding} = 1, \text{bias} = \text{False})$$
$$\rightarrow \text{BatchNorm2d}(C_{\text{out}}) \rightarrow \text{ReLU},$$
$$\text{Conv2d}(C_{\text{out}}, C_{\text{out}}, \text{kernel} = 3, \text{stride} = 1, \text{padding} = 1, \text{bias} = \text{False})$$
$$\rightarrow \text{BatchNorm2d}(C_{\text{out}}) \rightarrow \text{ReLU}.$$

   This construction replaces a single wide $7 \times 7$ convolution with a sequence of three $3 \times 3$ convolutions, which reduces parameters while preserving the effective receptive field and adds extra non-linearities.

2. **Spatial downsampling.** After each block we apply $2 \times 2$ max-pooling with stride 2 to halve the spatial resolution:

   - Block 1: $3 \rightarrow 64$ channels, then MaxPool2d(2). For $32 \times 32$ inputs this gives $16 \times 16 \times 64$.
   - Block 2: $64 \rightarrow 128$ channels, then MaxPool2d(2). Output: $8 \times 8 \times 128$.
   - Block 3: $128 \rightarrow 256$ channels, then MaxPool2d(2). Output: $4 \times 4 \times 256$.

3. **Channel projection and global pooling.** The final $4 \times 4 \times 256$ feature map is projected to a `flatten_size`-dimensional representation using a $1 \times 1$ convolution:

$$\text{Conv2d}(256, \texttt{flatten\_size}, \text{kernel} = 1, \text{bias} = \text{False})$$
$$\rightarrow \text{BatchNorm2d}(\texttt{flatten\_size}) \rightarrow \text{ReLU}.$$

   We then apply global average pooling over the spatial dimensions via AdaptiveAvgPool2d(1), producing a tensor of shape $(B, \texttt{flatten\_size}, 1, 1)$, which is flattened to a vector of length `flatten_size` per example.

In our experiments we test `flatten_size` $= 256$ and $= 512$, the latter of which is identical to the CLIP-embedding dimension. This representation $\mathbf{x}$ is then fed into the MLP classifier as before, $\mathbf{z} = \mathbf{W_2} \cdot \text{ReLU}(\mathbf{W_1}\mathbf{x} + \mathbf{b}_1) + \mathbf{b}_2$. In this setting, when testing both a `flatten_size` of 256 and of 512, FPE delivers improvements, especially in the interference-constrained settings (Figure 5c and d).

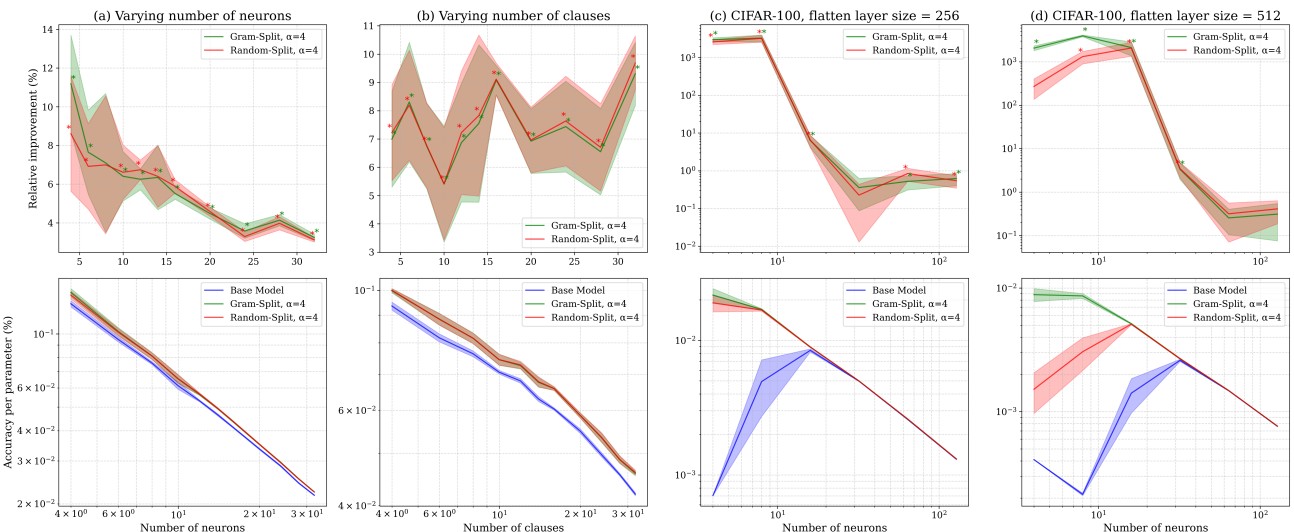

*Figure A5.* Jointly learning the features and classifying them is improved via increasing width. (a) Training a classifier on a Boolean DNF task with eight clauses of four literals each while varying the number of neurons and with an embedding layer. (b) Training a classifier on a Boolean DNF task with eight neurons while varying the number of clauses, each of which contains four literals, and with an embedding layer. (c) Training a CNN on CIFAR-100 with an embedding dimension of 256. (d) Training a CNN on CIFAR-100 with an embedding dimension of 512. The first row indicates the improvement in test accuracy of the FPE model relative to the dense baseline, for varying hidden dimensions. The second row shows the test accuracy per parameter for each configuration. In all figures, the number of neurons refers to the number of neurons pre-expansion. Relative improvement is calculated as before. Error bars indicate one standard error of the mean. * indicates $p < 0.05$. Results collected over ten trials.

The raw values that created Figure 5 are provided below.

*Table A8.* Training a classifier on a Boolean DNF task with eight clauses of four literals each while varying the number of neurons and with an embedding layer. Results averaged across 10 trials.

| Hidden size | Dense | Gram-Split | Random-Split |
|:---:|:---:|:---:|:---:|
| 4 | $70.3 \pm 2.1$ | $\mathbf{78.2 \pm 2.9}$ | $76.3 \pm 2.9$ |
| 6 | $75.2 \pm 1.8$ | $\mathbf{81.1 \pm 2.9}$ | $80.5 \pm 2.8$ |
| 8 | $80.2 \pm 1.1$ | $\mathbf{85.7 \pm 2.8}$ | $\mathbf{85.7 \pm 2.8}$ |
| 10 | $80.8 \pm 2.7$ | $86.2 \pm 3.4$ | $\mathbf{86.4 \pm 3.5}$ |
| 12 | $84.1 \pm 0.7$ | $89.3 \pm 0.9$ | $\mathbf{89.8 \pm 0.9}$ |
| 14 | $86.0 \pm 0.9$ | $91.3 \pm 0.7$ | $\mathbf{91.4 \pm 0.6}$ |
| 16 | $87.1 \pm 0.6$ | $91.9 \pm 0.6$ | $\mathbf{92.2 \pm 0.5}$ |
| 20 | $88.9 \pm 0.6$ | $92.8 \pm 0.5$ | $\mathbf{92.9 \pm 0.6}$ |
| 24 | $90.7 \pm 0.6$ | $\mathbf{93.9 \pm 0.4}$ | $93.6 \pm 0.5$ |
| 28 | $90.0 \pm 0.4$ | $\mathbf{93.7 \pm 0.3}$ | $93.6 \pm 0.3$ |
| 32 | $91.6 \pm 0.3$ | $\mathbf{94.6 \pm 0.3}$ | $94.4 \pm 0.3$ |

*Table A9.* Training a classifier on a Boolean DNF task with eight neurons while varying the number of clauses, each of which contains four literals, and with an embedding layer. Results averaged across 10 trials.

| Number of clauses | Dense | Gram-Split | Random-Split |
|---|---|---|---|
| 4 | $86.8 \pm 1.5$ | $92.7 \pm 0.8$ | $\mathbf{92.9 \pm 0.6}$ |
| 6 | $81.1 \pm 1.4$ | $\mathbf{87.8 \pm 2.2}$ | $87.7 \pm 2.2$ |
| 8 | $80.6 \pm 1.0$ | $\mathbf{86.2 \pm 2.0}$ | $\mathbf{86.2 \pm 1.9}$ |
| 10 | $79.1 \pm 0.6$ | $\mathbf{83.4 \pm 2.0}$ | $\mathbf{83.4 \pm 1.9}$ |
| 12 | $80.6 \pm 0.8$ | $86.0 \pm 1.2$ | $\mathbf{86.2 \pm 1.3}$ |
| 14 | $78.6 \pm 0.8$ | $84.4 \pm 1.7$ | $\mathbf{84.6 \pm 1.8}$ |
| 16 | $79.2 \pm 0.4$ | $\mathbf{86.4 \pm 0.6}$ | $\mathbf{86.4 \pm 0.7}$ |
| 20 | $79.1 \pm 0.7$ | $84.5 \pm 0.9$ | $\mathbf{84.6 \pm 0.9}$ |
| 24 | $77.9 \pm 0.7$ | $83.7 \pm 1.6$ | $\mathbf{83.9 \pm 1.6}$ |
| 28 | $77.4 \pm 0.4$ | $82.5 \pm 1.3$ | $\mathbf{82.6 \pm 1.3}$ |
| 32 | $76.6 \pm 0.5$ | $83.7 \pm 0.8$ | $\mathbf{84.0 \pm 0.8}$ |

*Table A10.* Training a CNN on CIFAR-100 with an embedding dimension of 256. Results averaged across 10 trials.

| Number of neurons | Dense | Gram-Split | Random-Split |
|---|---|---|---|
| 4 | $1.0 \pm 0.0$ | $\mathbf{30.9 \pm 3.6}$ | $27.0 \pm 3.8$ |
| 8 | $14.1 \pm 6.3$ | $\mathbf{48.4 \pm 0.7}$ | $47.9 \pm 0.9$ |
| 16 | $48.0 \pm 1.2$ | $50.8 \pm 0.4$ | $\mathbf{50.9 \pm 0.4}$ |
| 32 | $56.7 \pm 0.2$ | $\mathbf{56.9 \pm 0.2}$ | $\mathbf{56.9 \pm 0.2}$ |
| 64 | $58.9 \pm 0.1$ | $59.2 \pm 0.1$ | $\mathbf{59.4 \pm 0.2}$ |
| 128 | $59.7 \pm 0.1$ | $\mathbf{60.0 \pm 0.1}$ | $\mathbf{60.0 \pm 0.1}$ |

*Table A11.* Training a CNN on CIFAR-100 with an embedding dimension of 512. Results averaged across 10 trials.

| Number of neurons | Dense | Gram-Split | Random-Split |
|---|---|---|---|
| 4 | $1.0 \pm 0.0$ | $\mathbf{21.7 \pm 2.6}$ | $3.7 \pm 1.3$ |
| 8 | $1.0 \pm 0.0$ | $\mathbf{42.3 \pm 1.9}$ | $14.9 \pm 4.4$ |
| 16 | $13.8 \pm 4.3$ | $\mathbf{50.4 \pm 0.6}$ | $50.0 \pm 0.5$ |
| 32 | $50.9 \pm 1.0$ | $52.5 \pm 0.5$ | $\mathbf{52.6 \pm 0.5}$ |
| 64 | $57.9 \pm 0.2$ | $58.0 \pm 0.2$ | $\mathbf{58.1 \pm 0.2}$ |
| 128 | $59.7 \pm 0.1$ | $59.8 \pm 0.1$ | $\mathbf{59.9 \pm 0.1}$ |

## A.8. Validating Fixed Parameter Expansion at scale and in practical scenarios

### A.8.1. SCALING TO DEEPER ARCHITECTURES AND HIGH-DIMENSIONAL OUTPUTS

We first apply FPE to deeper (5, 7, and 9-layer) MLPs on CLIP embeddings of CIFAR-100, incorporating LayerNorm before each activation function for training stability. To scale FPE while maintaining a fixed parameter count, we use an alternating expansion strategy.

Concretely, for hidden layers $i$, $i + 1$, and $i + 2$, when we expand $i$ from $h \to \alpha h$ neurons, the subsequent layer $i + 1$ must also be adjusted to accept a wider input, from $h \to \alpha h$ as well. We then apply the "re-sparsify" method as described before (Section 2.2), and do not expand the neurons in layer $i + 1$. Thus, for hidden layer $i + 2$, we do not need to worry about expanding the input, and can simply expand the number of neurons again. We repeat this process with pairs of hidden layers, stopping before and not expanding the final classification head.

This alternating approach allows the network to benefit from the increased neuron capacity of FPE while maintaining stability and avoiding the excessive sparsity that could arise from expanding consecutive layers. As the results below show, FPE provides significant accuracy improvements, especially in narrower, more constrained models, confirming its compatibility with deeper, standard architectures (Table A12).

*Table A12.* Relative improvement in test accuracy from applying FPE to deeper MLPs with varying baseline hidden dimensions. Networks trained on CLIP embeddings of CIFAR-100. $\alpha = 2$. Results averaged across 30 trials.

| Depth | Hidden 4 | Hidden 8 | Hidden 12 | Hidden 16 |
|-------|----------|----------|-----------|-----------|
| 5 | 28% | 2.6% | 0.4% | 0.2% |
| 7 | 34% | 11% | 1.1% | 0.8% |
| 9 | 104% | 8.7% | 2.0% | 1.4% |

By not expanding the final classification layer, FPE can also be applied to tasks with a large number of classes. We validate this by applying FPE to a 5-layer MLP trained on the full ImageNet-1k dataset (Table A6), where it consistently improves performance over the dense baseline.

A.8.2. COMPATIBILITY WITH ADVANCED AND STRUCTURED SPARSITY

To further assess FPE's practical utility, we test its compatibility with two key techniques: dynamic sparse training and hardware-friendly structured sparsity.

Inspired by RigL (Evci et al., 2020), we investigate if FPE can be enhanced with a dynamic mask. After the initial FPE split, we periodically update the sparsity mask by randomly unmasking a fraction of weights and, to maintain a fixed parameter count, re-pruning an equal number of the lowest-magnitude weights. This learnable approach yields consistent gains over our one-shot FPE. This approach improves upon random one-shot FPE by 4.5% for 4 originally dense neurons, 1.9% for 8 originally dense neurons, and 0.5% for 16 originally dense neurons, confirming this is a promising direction for future work. This corresponds to relative improvements of 53%, 14%, and 3.0% over the respective dense models. These were collected over ten trials for the optimal rewiring hyperparameters on a single layer MLP trained on CIFAR-100 embeddings. This demonstrates that FPE provides a strong foundation that can be enhanced with more sophisticated, learnable sparse training techniques.

To explore FPE's potential for efficient inference, we constrain its masks to a 2:4 structured sparsity pattern, which is supported by modern hardware accelerators (Pool et al., 2021). For an expansion factor of $\alpha = 2$, each group of four consecutive weights in the original dense neuron is randomly assigned such that two weights connect to one sub-neuron and the other two connect to the second. As shown in Table A13, this structured approach achieves performance that is highly competitive with unstructured random sparsity. This is a promising result, as it suggests that FPE can be adapted to leverage significant hardware speedups without a meaningful performance trade-off.

*Table A13.* 2:4 structured sparsity FPE performs comparably to random sparsity FPE on CIFAR-100. RI denotes relative improvement. $h = 8, \alpha = 2$. Results averaged across 10 trials.

| Dense Accuracy | 2:4 Sparsity FPE Accuracy | Random Sparsity FPE Accuracy | 2:4 Sparsity FPE RI | Random Sparsity FPE RI |
|---|---|---|---|---|
| 53.89% | 61.67% | 61.92% | 14.42% | 14.89% |

A.8.3. STORAGE AND COMPUTE COSTS

**Storage and Memory** The number of non-zero weights, which constitutes model storage, is identical by construction. The expanded network additionally stores which sub-neuron each weight belongs to, but this requires only $\log_2 \alpha$ bits per weight, negligible relative to the weight values. The network does use more activation memory during forward passes, but this overhead is modest relative to the weight storage.

**Compute** Since the sparse mask is fixed after initialization and only modest fine-tuning is required after splitting, training overhead is likely limited. For inference, we provide initial evidence of hardware compatibility. Table A13 shows that FPE with 2:4 structured sparsity, natively supported by NVIDIA Ampere architecture (Pool et al., 2021), achieves performance comparable to unstructured FPE. Recent work on sparse matrix multiplication further suggests that practical efficiency of sparsification is rapidly improving (Macko & Boža, 2025).

A.8.4. COMPARISON OF FIXED PARAMETER EXPANSION TO DROPCONNECT

To compare our technique to DropConnect (Wan et al., 2013), we compare FPE to two DropConnect-style baselines that match the FPE setting as closely as possible while using standard DropConnect training and inference rules.

- a **non-disjoint FPE** baseline, which keeps the "one-shot" sparse architecture but removes the disjointness constraint on incoming weights to simulate a single DropConnect instantiation

- a **DropConnect** baseline in the standard sense, where a new weight mask is sampled at each forward pass during training and a dense, scaled weight matrix is used at test time.

Both baselines use the same warmup schedule and expansion factor as FPE and are designed so that the *expected* number of active parameters per forward pass during training matches the dense and FPE models. The key differences are (i) whether the mask is sampled once or resampled every step, and (ii) whether supports across duplicated neurons are disjoint.

**Non-disjoint FPE (one-shot random mask).** This baseline is architecturally closest to FPE and serves to isolate the effect of disjointness.

1. **Warmup (dense).** We first train the original dense classifier (with hidden width $r$) for the same 25 pre-training epochs as in our FPE experiments (Section A.3).

2. **Neuron duplication.** After warmup, we duplicate each hidden neuron $\alpha$ times, as in FPE. Let $\mathbf{W_1} \in \mathbb{R}^{h \times d}$ and $\mathbf{W_2} \in \mathbb{R}^{C \times h}$ denote the input and output weight matrices of the dense model. We construct duplicated matrices

$$\mathbf{W_1}' \in \mathbb{R}^{\alpha h \times d}, \quad \mathbf{W_2}' \in \mathbb{R}^{C \times \alpha h},$$

by copying each row of $W_1$ and each column of $W_2$ into $\alpha$ identical copies (i.e., we increase the number of neurons but do not yet introduce sparsity).

3. **Mask initialization** We then simulate how FPE would create a mask for each neuron in $\mathbf{W_1}$ and $\mathbf{W_2}$ via re-sparsification (Section 2.2), yielding $\mathbf{M_{1,FPE}}$ and $\mathbf{M_{2,FPE}}$. We keep $\mathbf{M_{2,FPE}}$ as $\mathbf{M_2}$ for DropConnect's $\mathbf{W_2}$, but $\mathbf{M_{1,FPE}}$ is not used: it is purely to set how many weights per neuron are non-zero in both DropConnect setups.

4. **One-shot random masking.** For each neuron $n_i$ in $\mathbf{W_1}$, we find how many non-zero weights there are in $\mathbf{M_{1,FPE}}$ for $n_i$. To construct $\mathbf{M_1}$, simply sample that many weights randomly once. This yields a sparse, widened architecture in which each duplicated neuron has the same degree as under FPE, but the supports of different neurons can overlap and some input dimensions may be dropped entirely. The mask $\mathbf{M_1}$ is sampled once after warmup and then kept fixed for the remainder of training and at test time.

5. **Training and inference.** We continue training the network with $\mathbf{W_1}' \odot \mathbf{M_1}$ as a fixed sparse weight matrix (only surviving weights are updated), and use the same fixed mask at inference. Thus this baseline is a one-shot, non-disjoint alternative to FPE that maintains a fixed number of non-zero parameters at test time but does not minimize feature collisions across neurons.

Empirically, this non-disjoint FPE baseline underperforms both true FPE and, at large expansion factors, even the dense model (Figure A6 and Table A14). This indicates that it is not widening alone that matters, but specifically that FPE's disjoint allocation of incoming weights reduces collisions and improves performance.

**DropConnect (weight dropout with dense test-time weights).** To more closely follow the standard DropConnect setting, we also implement a baseline with *resampled* masks during training and dense weights at test time.

1. **Warmup, duplication, and mask initialization.** As above, we first warm up a dense classifier, duplicate each hidden neuron $\alpha$ times, and initialize a sparsity mask to obtain $\mathbf{W_1}', \mathbf{W_2}', \mathbf{M_1}, \mathbf{M_2}$.

2. **DropConnect training.** During the continued training phase, we apply DropConnect to the incoming weights $\tilde{W}_1$ by sampling a new binary mask at each forward pass in the same way as before. For each neuron $n_i$ in $\mathbf{W_1}$, find how many non-zero weights there are in $\mathbf{M_{1,FPE}}$ for $n_i$. To construct $\mathbf{M_1}$, simply sample that many weights randomly per forward pass. This yields a sparse, widened architecture in which each duplicated neuron has the same degree as under FPE. After $\mathbf{M_1}$ is resampled for each forward pass during training, $\mathbf{W_1}' \odot \mathbf{M_1}$ is used as the sparse weights. For simplicity, we apply DropConnect only to $\mathbf{W_1}'$; applying it to $\mathbf{W_2}'$ does not change the qualitative conclusions.

3. **Inference with weight averaging.** At test time we follow the standard DropConnect inference rule (Wan et al., 2013): we do not sample a mask, but instead evaluate the network with dense, deterministically scaled weights

$$\mathbf{W_{1,TEST}} = p\mathbf{W_1}'$$

where $p$ is the number of non-zero weights in $\mathbf{M_1}$ divided by the total number of weights in $\mathbf{M_1}$, as per Wan et al. (2013). In practice, this is slightly less than $\frac{1}{\alpha}$ due to the re-sparsification procedure. Thus, unlike FPE and the non-disjoint FPE baseline, the DropConnect model is dense at inference and does not obey a fixed non-zero parameter budget; it serves as a regularization baseline rather than a fixed-parameter architectural comparator.

Intuitively, the non-disjoint FPE baseline tests whether widening with a single random sparse mask (but without disjointness) is sufficient to obtain FPE's benefits, while the DropConnect baseline tests whether repeated random masking of weights during training can account for the effect. In our CIFAR-100 CLIP experiments (Figure A6 and Table A15), both baselines underperform FPE: the non-disjoint FPE model performs worse than FPE and often worse than the dense model at large expansion factors, and the DropConnect model, despite having access to more active weights at test time (dense $p\mathbf{W_1}'$), still remains below both the dense model and FPE. These results support our claim that FPE's gains arise specifically from its structured, disjoint allocation of incoming weights that reduces feature collisions, rather than from dropout-style random masking alone.

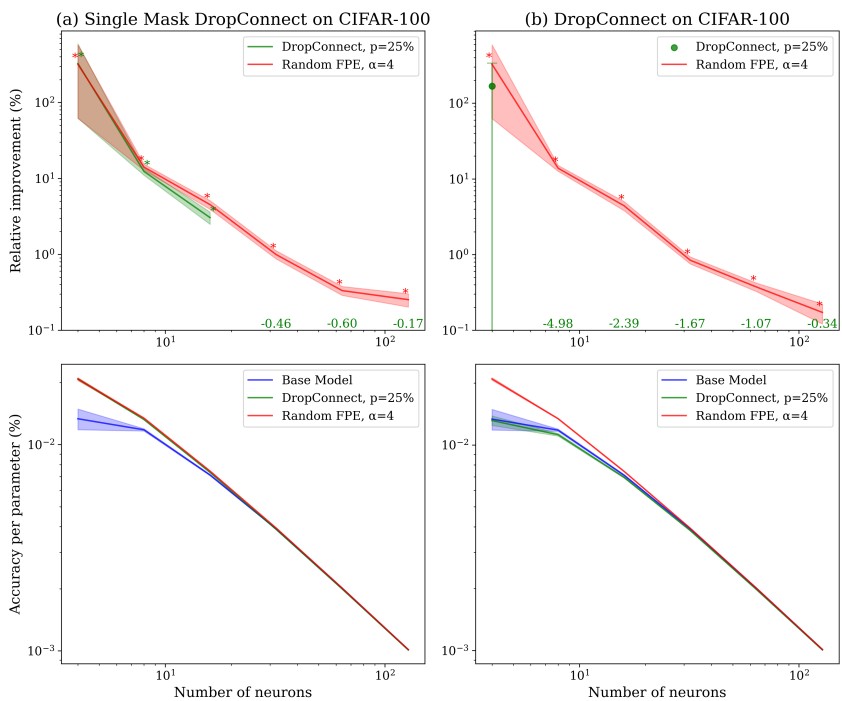

*Figure A6.* Comparison of FPE to DropConnect-style controls. (a) Single DropConnect mask instantiation on CLIP-embeddings of CIFAR-100, analogous to non-disjoint FPE. (b) DropConnect with random mask sampling during training and expected value weight scaling during evaluation. Relative improvement is calculated as before. Error bars indicate one standard error of the mean. * indicates $p < 0.05$. Results collected over ten trials.

*Table A14.* Single Mask DropConnect

| Number of neurons | Dense | Single Mask DC | Random-Split FPE (ours) |
|---|---|---|---|
| 4 | $32.7 \pm 3.8$ | $\mathbf{51.0 \pm 0.6}$ | $50.9 \pm 0.7$ |
| 8 | $57.8 \pm 0.7$ | $64.8 \pm 0.1$ | $\mathbf{65.8 \pm 0.2}$ |
| 16 | $69.7 \pm 0.5$ | $71.8 \pm 0.1$ | $\mathbf{72.8 \pm 0.1}$ |
| 32 | $76.4 \pm 0.1$ | $76.0 \pm 0.1$ | $\mathbf{77.1 \pm 0.1}$ |
| 64 | $78.6 \pm 0.1$ | $78.1 \pm 0.1$ | $\mathbf{78.8 \pm 0.1}$ |
| 128 | $79.2 \pm 0.0$ | $79.1 \pm 0.0$ | $\mathbf{79.4 \pm 0.0}$ |

*Table A15.* DropConnect

| Number of neurons | Dense | DropConnect | Random-Split FPE (ours) |
|---|---|---|---|
| 4 | $32.7 \pm 3.8$ | $32.2 \pm 1.7$ | $\mathbf{51.2 \pm 0.7}$ |
| 8 | $57.8 \pm 0.7$ | $54.9 \pm 0.8$ | $\mathbf{65.7 \pm 0.2}$ |
| 16 | $69.7 \pm 0.5$ | $68.0 \pm 0.5$ | $\mathbf{72.8 \pm 0.1}$ |
| 32 | $76.4 \pm 0.1$ | $75.1 \pm 0.1$ | $\mathbf{77.0 \pm 0.1}$ |
| 64 | $78.6 \pm 0.1$ | $77.7 \pm 0.1$ | $\mathbf{78.9 \pm 0.1}$ |
| 128 | $79.2 \pm 0.0$ | $78.9 \pm 0.1$ | $\mathbf{79.3 \pm 0.0}$ |

## A.9. Circuit Analysis and Semantics of Disentangled Features

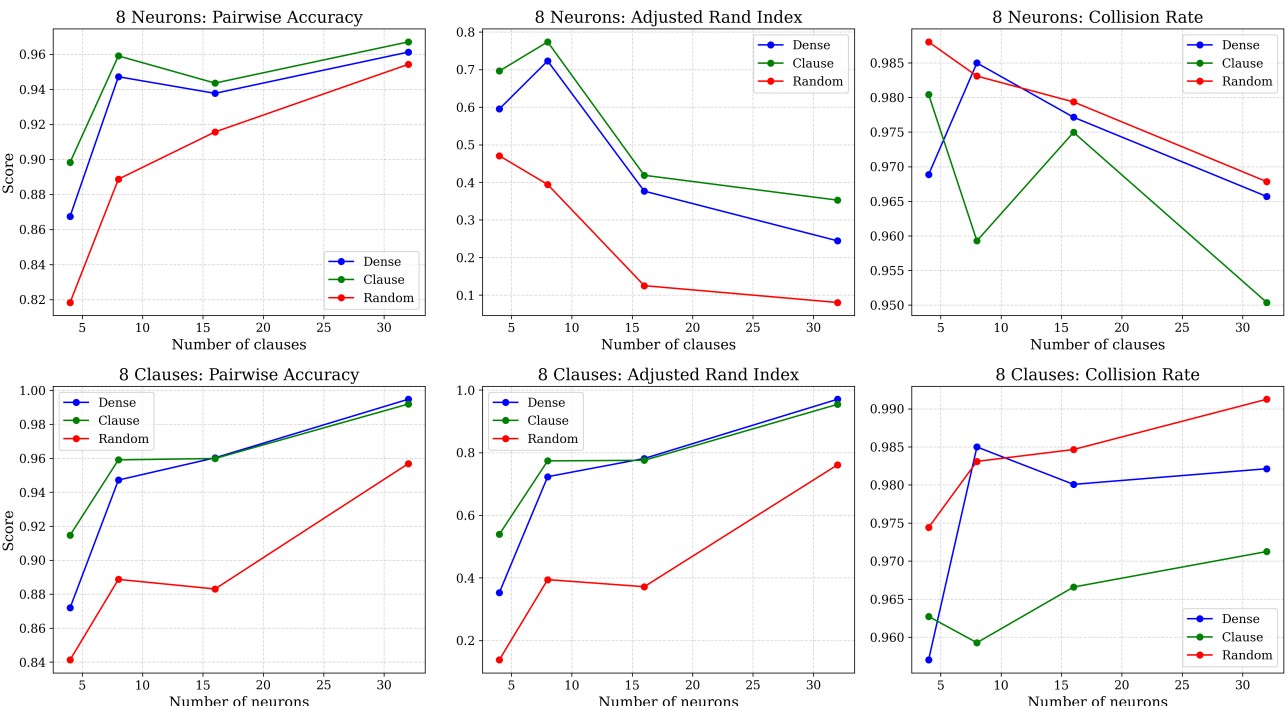

*Figure A7.* Sweep of model configurations evaluating three metrics: Pairwise Accuracy, Adjusted Rand Index (ARI), and Collision Rate. Top row: models with 8 neurons while varying the number of clauses. Bottom row: models with 8 clauses while varying the number of neurons. Each plot compares Dense, Clause-based, and Random connectivity patterns.

FPE is used in our work primarily as a tool for experimentally probing how width and fixed-parameter sparsity interact to reduce interference, not as a dedicated interpretability method. Our core thesis concerns the relationship between neuron count, collisions, and performance under a fixed non-zero parameter budget, rather than semantic or mechanistic interpretability. However, we recognize its connection to interpretability, and here explore preliminary mechanistic evidence to reveal reduced entanglement in clause-split architectures by conducting two additional empirical analyses: (1) an edge-level collision metric that directly measures circuit overlap, and (2) a semantic clustering analysis of hidden representations via Gram-matrix structure. Together, these analyses, though not intended as a full interpretability study, provide initial mechanistic and representational evidence that neuron expansion reduces interference and increases clause-specific structure.

### Experimental Setup for the Model Sweep

For the architectural sweeps shown in Figure A7, all values are averaged over 10 trials. Dense models were first trained for 25 epochs, after which their neurons were split. The resulting split models were then trained for an additional 25 epochs. This ensures that dense and split variants receive comparable optimization budgets.

For the sweeps, we explored two complementary axes of variation: (1) we fixed the number of clauses at 8 and varied the number of neurons, and (2) we fixed the number of neurons at 8 and varied the number of clauses. In all experiments, each clause contained 4 literals, ensuring consistent clause size across model families.

### Mechanistic Edge-Collision Analysis

We add an additional mechanistic edge-collision metric through circuit analysis, designed to trace how individual hidden-layer edges are shared across clause computations. For each test sample, we identify the clause that is satisfied and trace the hidden-layer edges that contribute to its activations. The collision rate counts the proportion of edges used by at least 2 distinct clauses. Lower collision rates correspond to cleaner circuit separation and reduced interference. As shown in Figure A7 (top–right and bottom–right panels), clause-split models generally achieve lower collision rates. This suggests that neuron expansion yields circuits with less overlap and more clause-specific routing.

**Clause-Aligned Representations via Gram-Matrix Clustering**

To additionally assess whether neuron expansion induces clause-aligned representations, we also cluster the hidden-layer Gram matrix using k-means and compare how well the resulting groups recover the ground-truth clause structure. We report two quantitative metrics:

- Pairwise Agreement: the fraction of literal pairs for which the model's cluster membership agrees with the true clause grouping.

- Adjusted Rand Index (ARI): a standard clustering-agreement score (ARI = 1 indicates perfect recovery).

Across all tested configurations, clause-split models show higher pairwise agreement and ARI than the unexpanded baseline, suggesting that expansion pushes features toward clause-aligned, semantically coherent groups. These improvements are visible in Figure A8, where the clustered Gram matrices form clearer block structures reflecting the underlying clause partition. Dense models also improve as neurons increase, but likely for a different reason: with enough neurons, even dense models have the capacity to represent all clause features fully, causing their performance to converge toward the clause-split models. However, at low and medium neuron counts, where there is greater superposition pressure, the clause-split architecture more consistently recovers clause-aligned structure. The Gram-matrix visualizations confirm this: clause-split models form clean block-diagonal clusters early, whereas dense models only approximate this structure once further parameterized.

Together, the collision analysis and the semantic clustering provide complementary initial evidence that neuron expansion reduces interference, improves clause alignment, and yields more interpretable hidden features. We view these mechanistic analyses as preliminary but promising evidence that neuron expansion reduces circuit-level interference. While the separation between dense and clause-split models is not extreme in all configurations, the trend is consistent across runs: clause-split models exhibit lower collision rates and higher clause-recovery scores. We include these analyses to illustrate the mechanism suggested by our theoretical model, and we view more comprehensive mechanistic studies, such as scaling these analyses to deeper or real-world networks, as an important direction for future work.

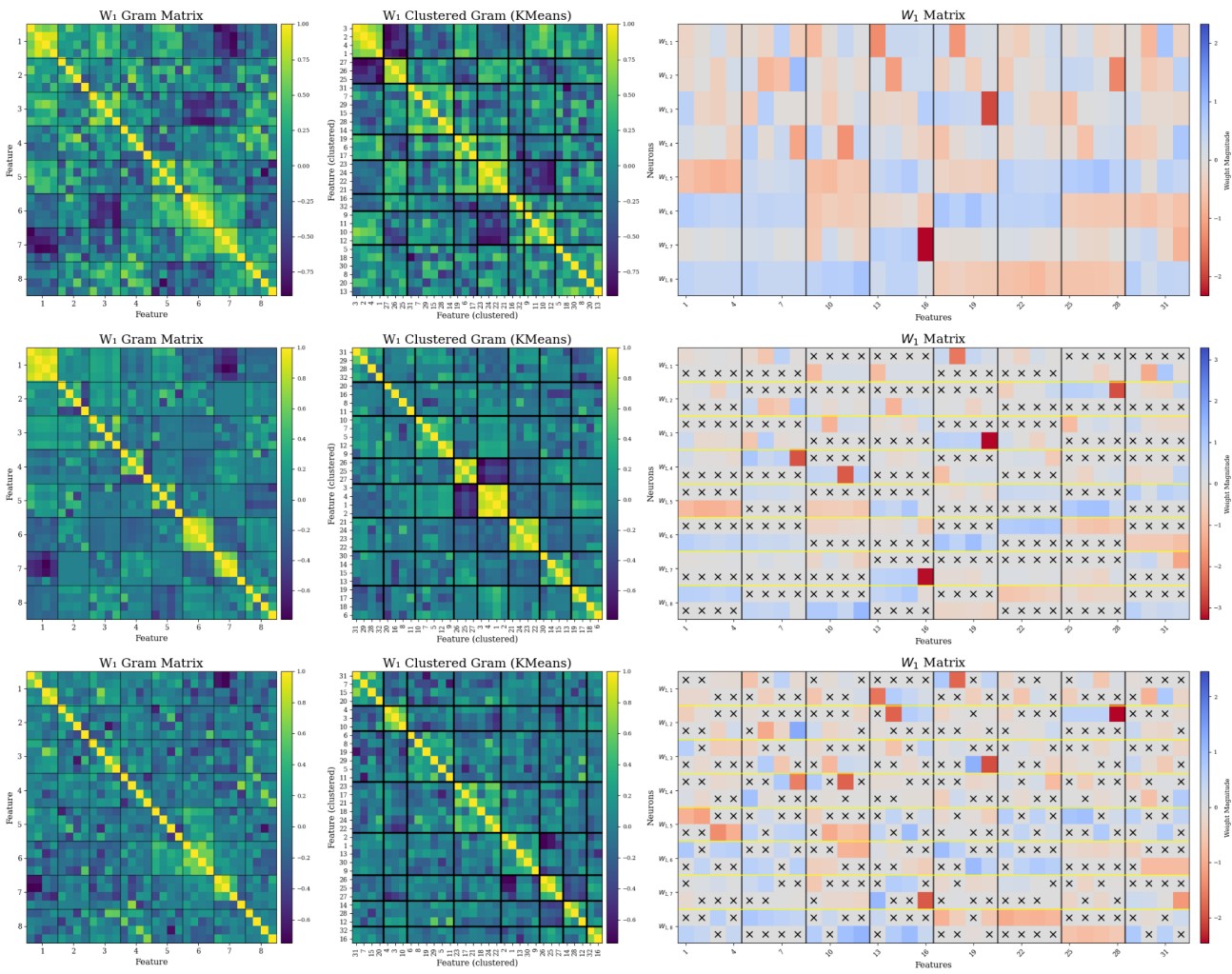

*Figure A8.* Gram matrix, clustered gram matrix (by k-means), and $W_1$ matrix for a boolean network of size: 32 literals, 4 literals per clause, and 8 neurons. Top row: dense model, middle row: clause-split model, bottom row: randomly-split model.

