# OpenReview forum: "Expand Neurons, Not Parameters"
_ICML.cc/2026/Conference — ICML 2026 regular_

### Official Review · Reviewer_CBaE · 2026-02-13

**Soundness:** 3
**Presentation:** 4
**Significance:** 3
**Originality:** 3
**Overall Recommendation:** 5
**Confidence:** 5

**Summary:**

This paper argues that, under a fixed parameter budget, it can be better to increase the number of neurons (width) rather than increase the number of learnable weights, by doing so via structured sparsification. The core claim is that widening while keeping the number of non-zero connections constant reduces feature interference / superposition—i.e., it becomes less likely that unrelated features are forced to share the same neuron—leading to improved accuracy and cleaner representations.

Methodologically, the authors propose Fixed-Parameter Expansion (FPE): train a dense network for a warmup period, then expand each hidden neuron into multiple “sub-neurons” and partition the original neuron’s incoming weights across these sub-neurons so that each sub-neuron is sparser and the overall non-zero parameter count is preserved. They study both structure-aware splitting (e.g., keeping inputs from the same clause together in synthetic Boolean tasks) and random splitting, and then continue training the expanded sparse model.

The paper provides (i) a theoretical analysis in a controlled Boolean DNF setting showing that widening with reduced fan-in can maintain coverage of target features while sharply reducing the probability of collisions between features, (ii) empirical results on synthetic Boolean tasks demonstrating large gains (especially with structure-aware splitting) along with supporting diagnostics (e.g., weight-geometry/Gram-matrix structure and interference-related metrics), and (iii) experiments on vision benchmarks (including MLPs on raw images and MLP heads on frozen CLIP embeddings, plus some jointly trained setups) showing that FPE can improve performance particularly when the baseline model is narrow. The authors also compare against related sparsity/regularization baselines (e.g., DropConnect-style masking and non-disjoint variants) to support the claim that disjoint partitioning plus neuron expansion is the key ingredient rather than sparsity alone.

**Compliance With Llm Reviewing Policy:**

Affirmed.

**Final Justification:**

I remain positive on this paper and am maintaining my original score and recommendation. Overall, I found the submission technically solid, clearly presented, and interesting in both its empirical and mechanistic aspects. The central idea—reallocating a fixed nonzero-parameter budget across more neurons via structured sparsification to reduce feature collisions/interference—is clear, testable, and well aligned with the proposed FPE intervention. I also thought the paper did a strong job of supporting this narrative through a combination of theory in a controlled Boolean setting, empirical results on synthetic and real tasks, and diagnostics that go beyond top-line accuracy.

Across the main evaluation dimensions, I weighed the paper positively for several reasons. On soundness, the paper presents a coherent hypothesis, introduces a simple and targeted method, and supports it with a useful combination of analysis and experiments. In particular, I found the Boolean DNF setting valuable because it provides a clean environment in which the collision-reduction mechanism can be stated and tested directly. On presentation, this is one of the stronger aspects of the paper: the narrative is easy to follow, the method is conceptually simple, and the figures/diagnostics make the proposed mechanism tangible. On originality and significance, I think the main contribution is not just another sparse-training recipe, but a distinct reframing: under a fixed nonzero budget, increasing neuron count by disjointly partitioning connections can improve both performance and representation quality. That is a useful perspective that I expect others to build on.

The rebuttal was helpful and increased my confidence in the paper, even though it did not fully eliminate all of my concerns. In particular, I appreciated the clarifications on split timing, the additional context on comparisons to non-disjoint sparse baselines and RigL-style updates, and the more careful positioning of the method as complementary to broader sparse-training approaches rather than universally superior to them. I also appreciated the authors’ acknowledgment that the mechanistic evidence is strongest in the Boolean setting and more suggestive on real data, and their willingness to qualify the practical efficiency claims around “fixed nonzeros” versus actual wall-clock cost. These clarifications made the contribution feel better framed and more appropriately delimited.

At the same time, some of my original concerns remain. The strongest mechanistic support for the superposition/interference explanation still comes from the stylized Boolean domain, and the causal link on real datasets is not yet as definitive. Similarly, while the evidence on timing robustness is encouraging, I still think a fuller treatment of warmup dependence and stronger controls would be valuable. I also continue to think that the paper should position itself more explicitly relative to the broader sparse-training literature and be clearer about the regime in which FPE helps most, as well as the distinction between fixed nonzeros and practical deployment efficiency.

These remaining issues, however, did not change my overall positive assessment. For me, they are best understood as limitations on scope and framing rather than as flaws that undermine the core contribution. The paper already provides a strong combination of idea, evidence, and interpretability, and the rebuttal reinforced my view that the authors understand the appropriate boundaries of their claims and are willing to state them clearly in revision. For those reasons, I am maintaining my original score and recommendation.

**Key Questions For Authors:**

1. How dependent are the gains on the dense warmup + expansion timing?

If gains persist without warmup and are stable across timings, it strengthens the claim that FPE’s benefit is structural (reduced collisions) rather than an optimization artifact or “reset.” If gains vanish without warmup or are highly timing-sensitive, I would view the method as less robust/general.

2. How does FPE compare to strong sparse-training baselines under the same nonzero budget?

If FPE outperforms or is complementary to these baselines, it significantly increases my confidence that the contribution is not already captured by standard sparse-training practice. If comparable or worse, the paper’s novelty/impact would need to be reframed more as a mechanistic analysis than a generally superior method.

3. Do you have direct evidence that FPE reduces polysemanticity/interference on real datasets (beyond the Boolean setting)?

Strong mechanistic evidence outside the toy domain would substantially strengthen the paper’s central claim about superposition reduction as the driver of improved performance. If the effect is not observable (or weak) on real tasks, I would treat the Boolean results as insightful but the broader explanatory claim as less supported.

**Limitations:**

- Clarify the regime of applicability: Explicitly discuss that improvements appear strongest for narrow models / high-interference regimes, and may diminish as baseline width increases; state what is currently unknown (e.g., transfer to transformers / large-scale pretraining).

- Acknowledge optimization and pipeline dependence: Note that FPE is applied after dense warmup, and that it is unclear how robust results are to expansion timing or training-from-scratch; recommend this as future work.

- Discuss practical deployment limitations: “Fixed nonzeros” does not necessarily imply fixed latency/energy; performance depends on sparse kernel support and hardware. Add a short discussion of when FPE might not yield wall-clock gains (or could even hurt due to increased activation/memory footprint).

**Strengths And Weaknesses:**

### Soundness

#### Strengths

- Clear, testable hypothesis (reduce feature collisions/superposition by expanding width under fixed nonzero budget) and a method (FPE) that directly targets it.


- Controlled Boolean DNF setting provides a clean environment where “features” are well-defined; theory (coverage vs. collisions under reduced fan-in) matches empirical behavior.


- Empirical evaluation goes beyond accuracy with supportive diagnostics (e.g., weight-geometry / Gram structure, similarity/interference proxies) that are consistent with the proposed mechanism.


- Helpful ablations against DropConnect-like and non-disjoint variants, supporting that disjoint partitioning + expansion matters rather than sparsity alone.


#### Weaknesses

- Theoretical guarantees rely on stylized assumptions (DNF/clauses, clean feature structure); it’s less clear how directly this maps to realistic learned features.

- Warmup-then-expand pipeline could confound causality: gains may depend on dense warmup dynamics or act like an optimization “reset.” Controls (train-from-scratch expanded-sparse; warmup length sweep) would strengthen claims.


- Baseline set could be expanded to stronger sparse-training approaches under the same nonzero budget (e.g., dynamic sparse training / pruning schedules), to ensure improvements aren’t already captured by established methods.


- “Fixed nonzeros” is not the same as “fixed cost” in practice; without wall-clock/throughput benchmarking, efficiency implications remain somewhat speculative.


- Benefits appear regime-dependent (largest for narrow networks); clearer characterization of when/why it helps would improve robustness of conclusions.

---

### Presentation

#### Strengths

- Coherent narrative: superposition motivation → FPE procedure → toy theory/experiments → extension to vision/embeddings → ablations.

- Boolean section is pedagogical and interpretable; figures/diagnostics make the mechanism tangible.

- Method is conceptually simple and (mostly) easy to reproduce.

#### Weaknesses
- Feature-based splitting on real datasets often resembles random splitting in effect; the paper could clarify whether this reflects limitations of the heuristic or the underlying assumption about recoverable feature structure.

- Reproducibility would benefit from more explicit details: exact expansion timing, mask construction, sparsity pattern specifics, warmup schedule, and tuning protocol.

- Stronger positioning relative to broader sparse-training literature (dynamic sparsity, pruning+finetune, etc.) would improve contextual clarity.

---

### Significance

#### Strengths

- Addresses a relevant problem: improving performance under parameter/nonzero constraints—important for sparsity-aware training and deployment.

- Potentially general recipe: random splitting still helps, suggesting applicability beyond hand-engineered structure.

- Connects efficiency-oriented architecture choices with representation quality (superposition/interference), which could influence follow-up work in sparse models and interpretability.

#### Weaknesses

- Practical impact may be concentrated in narrow/compute-constrained regimes; gains diminish as baseline width increases.

- Unclear transfer to modern large-scale architectures (e.g., transformers) and settings (pretraining, long-context), limiting immediate broad applicability.

- Lack of strong real-hardware efficiency evidence makes it harder to assess significance for deployment-focused audiences.

---

### Originality

#### Strengths

- Novel framing and intervention: widen by disjointly partitioning weights to keep nonzeros fixed, motivated by superposition/feature-collision reduction.

- Provides a concrete collision-based explanation (and supporting diagnostics) rather than only reporting sparse/wide empirical improvements.

- Useful reframing: representational capacity can increase via neuron count even when parameter count does not.

#### Weaknesses

- Close adjacency to existing ideas (widen + prune/sparsify, sparse training/pruning pipelines); novelty rests heavily on the specific disjoint partitioning and superposition-based argument.

- Mechanistic originality is strongest in the toy Boolean domain; in realistic tasks, the evidence is more empirical and less explanatory, which may make the contribution feel more incremental unless further analysis tightens the link.

---

> ### Author Rebuttal · Authors · 2026-03-31
>
> We thank the reviewer for your detailed and supportive review. We especially appreciate your recognition of the paper's central hypothesis, the controlled Boolean setting, and the mechanistic analyses supporting the interference-reduction interpretation.
>
> We address your key questions first, then clarify the remaining points you raised.
>
> ## Response to key questions
> ### 1. Dependence on warmup and timing
> Our current evidence suggests that the effect is not especially timing-dependent. Figure 6b investigates this by varying when the split occurs. In all conditions, FPE produces significant improvement over the dense baseline after fine-tuning, regardless of when the split is applied. Earlier splitting does yield modestly better final accuracy, likely because the network has more training time to specialize the sub-neurons, but even late splits consistently outperform the dense baseline under the same total training budget. We view this as evidence that FPE's benefit is structural rather than an optimization artifact. At the same time, we agree that additional experiments would strengthen this point further, and we will make that limitation and future direction more explicit.
> ### 2. Comparison to sparse-training baselines
> We provide two relevant comparisons in the appendix. First, appendix A.8.3 compares FPE against both a non-disjoint sparse baseline and standard DropConnect (Wan et al., 2013) under matched conditions, and FPE outperforms both. Notably, the non-disjoint baseline, which has the same width and parameter count but allows overlapping sub-neuron supports, performs worse than FPE and sometimes worse than the dense model, isolating disjoint partitioning as a key ingredient. Second, appendix A.8.2 shows that FPE is complementary to dynamic sparse training, as applying RigL-inspired (Evci et al., 2020) mask updates on top of FPE yields further gain. We therefore view the current evidence as supporting two points: the gains are not captured by dropout-style random masking alone, and FPE is complementary to more sophisticated sparse-training ideas rather than a replacement for them. We will clarify this positioning more explicitly in the revised version.
> ### 3. Evidence of polysemanticity reduction in real datasets?
> Yes, although we agree it is less definitive than in the Boolean setting. On real tasks, the strongest direct evidence we currently provide is on CIFAR-100, where randomly split FPE models show lower mean cosine similarity between neuron weights across widths than dense baselines, consistent with reduced interference (Figure 6c). More broadly, in the real-data experiments we find that gains are largest in the high superposition pressure regime, which matches the paper’s central hypothesis. We agree that stronger feature-level diagnostics on realistic datasets would be valuable future work, and we will state that more clearly.
>
> ## Response to remaining points
> ### Soundness
> - Theoretical assumptions vs. realistic features: Our theoretical analysis intentionally uses a Boolean setting where features are well-defined, in order to cleanly isolate the collision-reduction mechanism. The consistent empirical improvements on real datasets, where no such clean structure exists, suggest the principle generalizes beyond the theoretical assumptions. We will clarify this distinction in the revision.
> - Regime dependence: We agree and will expand the limitations section. We note that regime-dependence is itself a prediction of superposition - FPE helps most when interference is high - which we view as a strength of the explanatory framework.
> ### Presentation
> - Feature-based ≈ random splitting: We believe this reflects limitations of the Gram-matrix clustering heuristic rather than the underlying assumption about recoverable feature structure. The heuristic uses k-means on the Gram matrix, which may not capture the full feature structure. More sophisticated methods, such as sparse autoencoders, could potentially recover better groupings. We will clarify this.
> - Reproducibility details: The full training and hyperparameter details are provided in appendix sections A.2, A.3, and A.8. We will make pointers more prominent in the main text.
> ### Significance
> - Practical deployment: We agree that fixed nonzeros does not automatically imply fixed wall-clock time or energy, and will add a discussion of this distinction. The 2:4 structured sparsity results (Table A12) provide initial hardware compatibility evidence, but further benchmarking is needed for practical deployment.
> ### Originality
> - Adjacency to prior work: Our intended claim is not that FPE is superior to prior techniques, but that it offers a distinct and interpretable intervention: reallocating a fixed non-zero budget across more neurons reduces collisions and improves performance. We will strengthen the positioning relative to prior sparse-training and pruning pipelines accordingly.
>
> Thank you again for the careful reading and constructive suggestions!

---

> > ### Author Rebuttal · Reviewer_CBaE · 2026-04-01
> >
> > Thank you for the detailed and thoughtful rebuttal. I selected **(c)** because the response is helpful and addresses several of my questions, but my remaining concerns are still only **partially resolved**, and they relate to the paper’s **core explanatory claim** rather than to minor presentation details.
> >
> > In particular:
> >
> > 1. **Warmup / timing dependence is clarified, but not fully closed.**
> >    The additional pointer to Figure 6b is useful, and the evidence that FPE helps across multiple split timings is encouraging. However, my original concern was broader: whether the observed gains are fundamentally due to the structural effect of reduced collisions, as opposed to depending on the specific warmup-then-expand pipeline. The rebuttal makes this concern weaker, but I still think a more complete answer would require stronger controls (for example, training-from-scratch expanded sparse variants, or a more systematic analysis of warmup dependence) than can be fully established in a short rebuttal.
> >
> > 2. **Comparison to stronger sparse-training baselines is improved, but the positioning still needs a fuller treatment.**
> >    The appendix comparisons to non-disjoint sparse baselines, DropConnect, and RigL-style updates are valuable, and I appreciate the clarification that the intended claim is complementarity rather than universal superiority. Still, this issue is central to how the contribution should be interpreted relative to the broader sparse-training literature. I think the paper would benefit from a more substantial revision that makes this positioning explicit and carefully delimits what is being claimed.
> >
> > 3. **The mechanistic claim is strongest in the Boolean setting, and remains less definitive on real data.**
> >    I appreciate the pointer to reduced cosine similarity on CIFAR-100 and the acknowledgment that the evidence is less definitive outside the toy setting. This is exactly why I selected (c): the paper’s central narrative is that reducing superposition/interference is the driver of the gains, but on realistic datasets this causal/mechanistic link is still suggestive rather than fully demonstrated. Strengthening that connection would likely require additional analysis beyond what can reasonably be added in rebuttal.
> >
> > 4. **The regime of applicability and practical implications still need fuller framing.**
> >    I agree with the authors that regime dependence can itself be consistent with the superposition explanation, and I appreciate the planned expansion of the limitations discussion. At the same time, questions about when FPE helps most, and how “fixed nonzeros” translates (or does not translate) to practical efficiency, remain important for understanding the paper’s broader significance.
> >
> > Overall, I found the rebuttal constructive and helpful, and it increased my confidence in several parts of the paper. My choice of **(c)** reflects that the remaining issues are substantive and would be best addressed through a **more complete revision of the manuscript**, rather than through short rebuttal clarifications alone.

---

> > > ### Author Response · Authors · 2026-04-05
> > >
> > > Thank you for the helpful acknowledgement, and for clarifying your remaining concerns. We agree that these concerns are important to address via a more careful framing in the revised manuscript. In the revision, we will explicitly qualify and clarify our claims along the four dimensions you identified:
> > >
> > > 1. Current results support robustness across split timings within the dense training, split, then finetuning pipeline, but do not fully rule out effects of optimization path
> > > 2. Our intended positioning relative to sparse training baselines is complementarity and focuses on mechanistic distinctiveness, not necessarily universal superiority
> > > 3. Our mechanistic evidence is strongest in the Boolean setting, where we have precise control over the features, while on real data it is suggestive rather than definitive
> > > 4. The gains appear strongest in high interference regimes, whether due to small network size or high feature demand, and fixed nonzeros do not imply fixed wall clock efficiency
> > >
> > > We appreciate your supportive review, and we are glad that the rebuttal increased your confidence in several parts of the paper. We are fully committed to incorporating these clarifications and delimitations in the revision.

---

### Official Review · Reviewer_Pffo · 2026-03-02

**Soundness:** 3
**Presentation:** 1
**Significance:** 3
**Originality:** 2
**Overall Recommendation:** 4
**Confidence:** 3

**Summary:**

The paper argues, based on a combination of theory and experiment, that increasing the number of neurons of a neural network while keeping the number of (non-zero) parameters constant improves performance. The idea is motivated by two distinct hypotheses: (i) the superposition hypothesis, which states that networks are forced to represent more features than they have neurons, and (ii) the lottery-ticket hypothesis, which suggests that sparse subnetworks can compete with or even beat their dense counterparts.

On a Boolean symbolic task, it is shown that that sparsifying neurons with fixed parameters preserves clause coverage while reducing feature collisions. Empirical analyses on the same task show that the benefits hold for random neuron splitting and, consistent with the theory, are highest for high polysemanticity. Finally, experiments with image classifiers over both fixed and learned embeddings also suggest that structured and random network widening at constant parameters leads to performance gains, again at a relatively small number of neurons where polysemantic load is high.

**Compliance With Llm Reviewing Policy:**

Affirmed.

**Final Justification:**

I chose "weak accept" (4) because, following the rebuttal, I think overall the paper is sound, fairly original, potentially impactful, but poorly presented. My concerns were related to the implications of the findings for large-scale models and the poor presentation of the paper. The authors committed to revising the paper's structure for clarity and provided clarifications and additional experiments suggesting that the findings transfer at large scale. For this reason, I raised my initial score (3) to a 4, but cannot recommend higher acceptance given the non-trivial revision needed.

**Key Questions For Authors:**

Please the major weaknesses above for my main questions and comments. I would be happy to raise my score if these are all properly addressed. I would also appreciate the authors' thoughts on the following questions:
* Is width the only possible axis along which the number of neurons can be increased at constant number of parameters? Is this not possible for depth?
* Do the authors have any insights as to why the benefits of FPE seem to be much larger for CIFAR-100 and ImageNet-100 (Figure 4)?
* What relationship or implications, if any, do the results have for mixture of experts models?
* Can the authors also comment on the relationship between their findings and those of Liu et al. (2025), who find that LLMs tend to operate in the strong-superposition regime? Given their large model width, shouldn't modern LLMs have low superposition and polysemanticity?

**References**

Liu, Y., Liu, Z., & Gore, J. (2025). Superposition yields robust neural scaling. arXiv preprint arXiv:2505.10465.

**Limitations:**

The authors clearly acknowledge the main limitation of their work in the introduction, in terms of the scope of their claims. However, as noted in the major weaknesses above, discussion of other limitations would also be useful for understanding how the work could be built upon.

**Strengths And Weaknesses:**

**Strengths**
* Clear motivation and interesting research question combining different literatures.
* Overall, clear and detailed explanations of results.
* Fairly good description of setup and experiments, with relevant pointers to the appendix for more details.
* Fair combination of theory and experiment, including an analysis of an analytically tractable toy model with experiments on more complex models and tasks.
* Good acknowledgment of limited scope of the claims.

**Weaknesses**

*Major*
* **Unclear contributions**: While the main idea and execution of experiments testing the idea are clear, the paper would benefit from a clearly structured set of contributions (e.g. as bullet points), especially given its close relationship to previous works. In addition, as the authors write, “as the network widens and has more capacity, the gains from expansion diminish due to decreasing superposition”. Given this, could the authors clarify the impact of the results for large-scale models (or even the brain) which are very wide?
* **Confusing structure/presentation of methods and results**: While individual results are often clearly explained in detail, the entire structure of the methods and results is hard to follow. For example, the neuronal expansion procedure is described in Section 2.5 after being first applied in Section 2.4. The theoretical results could also have been included in a separate section rather than in the methods. Related since many of the experiments are based on the Boolean symbolic tasks, it might have been more clear to separate analyses (theoretical or empirical) of this task with the other experiments on classifications tasks, following the structure suggested by the abstract. Adding a paragraph in the introduction outlining the structure of the paper would also be helpful.
* **No discussion of limitations or future directions**: While the authors clearly acknowledge the limited scope of their claims - and this is appreciated - it would also be useful if other limitations of the work would be discussed, to help the reader better understand how the work could be built upon.

*Minor*
* Many sentences in the paper end with a few words on a new line, so space could be better optimised. Related, the appendix puts each section on a new page, which feels unnecessary.
* The acronyms DNF and and FPE are not defined where they are first used, in Sections 2.1 and 2.3.2 respectively.
* Typo in lines 185-6: "is prepended" instead of "in prepended".
* The caption of Figure 2c should refer to left and right instead of top and bottom. Adding a label to the colorbar would also improve the clarity of these plots.
* All figures have very small fonts which makes them hard to read.
* There seems to be a missing reference to Figure 6 in the main text.

---

> ### Author Rebuttal · Authors · 2026-03-31
>
> We thank the reviewer for your thoughtful analysis. We are glad that the motivation, combination of theory and experiment, and acknowledgment of scope were appreciated. We address the major weaknesses and key questions below:
>
> ## Response to weaknesses
> ### 1. Clarifying contributions
> We will revise the introduction to include this structured summary of contributions:
> 1. We show that redistributing a fixed non-zero parameter budget across more neurons reduces feature interference and improves accuracy, emphasizing neuron count as an axis of model performance distinct from parameter count.
> 2. We provide a theoretical analysis showing that expanding width at fixed parameter count preserves feature coverage while reducing expected collisions, even under random weight partitioning.
> 3. We validate these predictions across symbolic Boolean tasks and real-world vision benchmarks, with gains largest in high-interference regimes, as predicted by the superposition framework.
> 4. We provide direct mechanistic evidence linking width expansion to reduced polysemanticity: feature capacity increases, cosine similarity decreases, and both strongly correlate with accuracy gains.
>
> Regarding larger models, while gains diminish as width increases under a fixed feature set, this does not imply irrelevance at scale. Recent works suggest the key quantity is the ratio of features to neurons. Liu et al. (2025) shows that LLMs remain in strong superposition even at large widths, because feature count grows with model capacity and task complexity. Theoretical work confirms this, showing that representable features grow with width (Adler & Shavit, 2024). Empirically, widespread superposition has been observed in production-scale models such as Claude 3 Sonnet (Templeton et al., 2024). We will add a discussion of these implications.
> ### 2. Improving structure and presentation
> We will reorganize without changing core content as follows:
> 1. Introduction (motivation, contributions, overview)
> 2. Method (architecture, FPE procedure, tasks)
> 3. Theoretical Analysis
> 4. Symbolic Experiments (case study, scaling, interference-performance relationship)
> 5. Real-World Generalization (frozen embeddings, joint learning, CIFAR-100 analysis)
> 6. Related Work
> 7. Conclusion and Discussion (summary, limitations, future work)
>
> This addresses the concerns by introducing FPE first, separating theory from results, and distinguishing Boolean from real-world experiments. We will also add a roadmap paragraph in the introduction.
> ### 3. Limitations and future work
> We will add an explicit discussion covering our focus on small/controlled models without direct applicability claims for transformers, improving neuron partitioning algorithms, and practical deployment costs.
>
> We will also address the sentence and appendix spacing, acronyms, typos, and figure captions and references in the revision.
>
> ## Response to key questions
> 1. Width vs. depth: Both axes affect how interference is distributed, but width is more natural for directly reducing it after training. FPE partitions incoming connections per neuron, targeting collisions that drive superposition. Extending this to depth is less straightforward, as splitting layers requires redistributing parameters across sequential computations. Depth-wise analogues remain an interesting future direction.
> 2. Larger gains on some tasks: The smaller gains on ImageNet-1k are likely partly architectural. Unlike the other tasks, it uses a deeper 5-layer MLP with alternating expansion, distributing interference across layers and reducing FPE's per-layer effect size. CIFAR-100 and ImageNet-100 compress 512-D embeddings through a narrow single-layer bottleneck into 100 classes, thus likely concentrating superposition pressure, making FPE more effective.
> 3. Relation to MoE: Both FPE and MoE reduce feature interference, but through complementary mechanisms. MoE dynamically routes inputs to subnetworks, increasing total parameters while keeping per-inference compute manageable. FPE partitions weights under a non-zero parameter budget. This makes FPE a cleaner setting for isolating the effect of parameter allocation across neurons from the confounding effect of increased total parameter count.
> 4. Relation to Liu et al.: Building upon our discussion above, our observation that width reduces superposition is complementary with Liu et al. Our controlled experiments hold feature count fixed while growing width, artificially relieving pressure. In realistic settings, that pressure is continually maintained by expanding feature demands. We view Liu et al.'s findings as directly motivating FPE at scale, since the geometric interference they identify as governing scaling behavior is what FPE targets. We will discuss this explicitly.
>
> Thank you again for the constructive engagement. We believe the revisions will substantially improve readability, and hope that these clarifications resolve your concerns.

---

> > ### Author Rebuttal · Reviewer_Pffo · 2026-04-01
> >
> > I thank the authors for the detailed response. I chose **(b)** because the rebuttal addresses most of my concerns. In particular, I appreciate:
> > * the explicit statement of contributions, which now makes the impact of the paper much more clear,
> > * the proposed revision of the paper structure, and
> > * the commitment to discussing limitations and future directions.
> >
> > The comparison with MoEs is interesting (in that parameter count is not controlled) and could be useful to add to the discussion or even introduction, to highlight the distinctiveness of the method proposed.
> >
> > I have one remaining question related to larger models. Given the results of Liu et al. (2025), and the recognition from the authors that in realistic settings superposition does not decrease at scale because of "expanding feature demands", then wouldn't one expect higher gains on ImageNet compared to CIFAR-10, at larger model size - which is the opposite of what the authors report? More broadly, isn't there a synthetic task where the number of features grows with the model size, complementary to the controlled setting considered by the authors? Given the promising results of Liu et al. (2025), I feel like that even a toy demonstration of the benefits of FPE with larger model size would greatly strengthen the paper's impact.

---

> > > ### Author Response · Authors · 2026-04-05
> > >
> > > Thank you for your response, and we are glad that the revisions motivated by your feedback, particularly the restructuring and clearer statement of contributions, have helped improve the clarity and impact of our paper. We also appreciate your suggestion regarding comparisons with MoEs, and will highlight this more prominently, either in the introduction or discussion.
> > >
> > > Regarding your point on how superposition scales with larger models, we agree that this is a crucial question. First, in Boolean networks, where we have exact controls of both the feature count (via clause number) and the network size (via neuron count), we directly investigate this scaling behavior in our original submission. In Figure 2c, we jointly vary clause and neuron count, and plot the relative improvement that FPE imparts over the dense baseline model. The key pattern is that when network size increases while feature demand increases alongside it, the gains from FPE persist along the diagonal of Figure 2c, rather than diminish. Moreover, at any fixed network size, increasing the number of features consistently increases the benefit of FPE, suggesting greater superposition pressure. We view this as controlled evidence for your broader point that larger models do not necessarily experience less superposition if the number of features they must represent grows as well. We agree this connection could have been made clearer in our earlier rebuttal.
> > >
> > > We recognize that this feature variation axis was missing in our experiments on real world datasets, and we appreciate your suggestion to study a setting where feature complexity grows with model size. In response, we have designed an additional experiment to more directly probe this in a realistic setting.
> > >
> > > In CIFAR-100, we vary the effective number of features by varying the number of classes that a network must classify, while simultaneously scaling model width. This serves as a practical proxy for increasing feature demands with model size, analogous to the previous Boolean setting. We then measure the relative improvement of FPE over dense models across these settings.
> > >
> > >
> > > For both split types (random and gram-matrix based), the expansion factor is 4, results are averaged over 10 trials, and * indicates p < 0.05.
> > >
> > > Split type: gram
> > >
> > > ||Num Classes|40|60|80|100|
> > > |-|-|-|-|-|-|
> > > |Hidden Size|||||
> > > |10||1.17±0.30*|4.38±0.57*|6.20±1.37*|9.53±1.80*|
> > > |15||0.83±0.29*|3.19±1.07*|2.64±0.36*|3.68±0.35*|
> > > |20||0.49±0.07*|0.96±0.14*|1.55±0.25*|1.80±0.15*|
> > > |25||0.02±0.11|0.45±0.06*|0.89±0.12*|1.05±0.15*|
> > >
> > > Split type: random
> > >
> > > ||Num Classes|40|60|80|100|
> > > |-|-|-|-|-|-|
> > > |Hidden Size|||||
> > > |10||1.07±0.26*|4.64±0.58*|6.52±1.41*|9.67±1.72*|
> > > |15||0.82±0.24*|3.17±0.97*|2.51±0.44*|3.62±0.33*|
> > > |20||0.29±0.05*|0.83±0.19*|1.68±0.21*|2.00±0.16*|
> > > |25||0.19±0.15|0.63±0.08*|0.80±0.14*|1.23±0.17*|
> > >
> > > This experiment shows the same qualitative pattern as in the Boolean setting. We find that when feature demand increases alongside model size, the gains from FPE are maintained much more strongly than when feature demands are kept constant, as in Figure 4b. In particular, at fixed width, increasing the number of classes consistently leads to larger improvements from FPE, which is what one would expect if superposition pressure is increasing. This supports the interpretation that the key quantity is not model size alone, but the ratio between feature demand and available neurons. We acknowledge that this experiment is only a proxy, since classes are an imperfect measure of underlying features, but it provides initial evidence in a more realistic setting that complements the controlled Boolean results. We will include this analysis in the revision and more clearly emphasize that FPE’s benefits are governed by superposition pressure rather than scale alone, aligning with the perspectives of Liu et al. (2025) and Adler & Shavit (2024).
> > >
> > > Finally, we note that directly comparing our results on ImageNet-1k and CIFAR-100 introduces several confounding factors, including differences in data distribution, feature complexity, and classifier architecture, making it difficult to isolate the effect of feature scaling alone. For this reason, we believe the matched Boolean and CIFAR-100 analyses provide a cleaner way to isolate the effect you raised.
> > >
> > > Thank you again for your careful and in depth engagement with our work. Your question and suggestion helped us clarify an important point and motivated an additional experiment that strengthens the paper framing and impact.

---

### Official Review · Reviewer_GceV · 2026-03-12

**Soundness:** 3
**Presentation:** 2
**Significance:** 3
**Originality:** 3
**Overall Recommendation:** 4
**Confidence:** 2

**Summary:**

This paper shows that increasing the number of neurons in a DNN without increasing the number of non-zero parameters can significantly improve model performance. The study claims that it is due to a reduction in interference between features that would originally share the same neuron. Experimental results show that splitting each neuron into sparser sub-neurons systematically reduces polysemanticity metrics in Boolean tasks, thereby enhancing task accuracy. Moreover, this conclusion is also replicated in real-world models such as classifiers based on CLIP embeddings, convolutional neural networks (CNNs), and deeper multilayer networks.

**Compliance With Llm Reviewing Policy:**

Affirmed.

**Final Justification:**

The author addressed my core concern. I recommend accepting this paper.

**Key Questions For Authors:**

1. If I understand correctly (please correct me if I'm wrong), the main conclusion of the paper is that increasing the number of neurons in a DNN, without increasing the number of non-zero parameters can improve model performance
- In my opinion, the new DNN has a much **larger solution space** compared to the original DNN because it contains more neurons, as long as the same number of non-zero parameters is maintained. In this sense, the conclusion is not surprising.  How do the authors view this question？
- The paper claims "improving performance without increasing the number of non-zero parameters. Such a direction is well-matched to modern accelerators, where memory movement of non-zero parameters…” However, does the new network require more storage due to the increased number of neurons? Also, does the larger scale of the network lead to significantly higher training costs compared to previous models? If so, the practical value of this approach may be limited. It would help if you can give some practical examples.

2. The statements "splitting each neuron into sparser sub-neurons" and "without increasing the number of non-zero parameters" seem contradictory in intuitive. Could you provide further explanation to clarify this?

**Limitations:**

Yes

**Strengths And Weaknesses:**

Strength:
1. The paper proposes an interesting claim: increasing the number of neurons in a DNN, without increasing the number of non-zero parameters, can improve model performance.

2. The motivation for the study is solid, which is supported by extensive previous work in the field.

Weakness：
1. The paper focuses on relatively small-scale networks rather than large models commonly used today. It remains uncertain whether the conclusions still work in large-scale models.

2. The writing is a little heavy and dense, which makes the paper challenging to read.

3. See Key Questions for further clarification.

---

> ### Author Rebuttal · Authors · 2026-03-31
>
> We would like to thank the reviewer for their positive assessment and insightful feedback. We are glad that you found the claim interesting and the motivation solid. Below, we address each of your key questions, then respond to the noted weaknesses.
>
> ## Response to Key Questions
> ### 1. Solution space increase
> We appreciate this subtle question. We do not view FPE as simply enlarging the solution space. Because FPE preserves the non-zero budget while imposing fixed sparsity and disjoint-support constraints, it does not add unconstrained degrees of freedom, but reallocates the same weight budget across more neurons. We therefore view FPE as altering, not necessarily enlarging, the hypothesis class in a way that reduces interference.
> The paper includes controls that directly test this. In appendix A.8.3, we compare FPE against a non-disjoint baseline that has exactly the same number of sub-neurons and non-zero parameters, but allows sub-neurons' inputs to overlap rather than being disjoint. If the full explanation were that more neurons yields a greater solution space, this baseline should perform comparably to FPE. Instead, it performs worse than FPE and sometimes worse than the dense model (Table A13). This indicates that the disjoint partitioning, which explicitly reduces feature collisions, is a key ingredient, not merely having more neurons.
> Literature also generally correlates more neurons with more parameters (e.g. Kaplan et al., 2020). Our contribution is to separate these two axes and emphasize how neuron count matters independently of parameter count, as well as show how increasing neurons reduces interference and improves accuracy.
> We will highlight this distinction in the discussion.
> ### 2. Storage and compute costs
> This is a valid concern, and we clarify each aspect:
>
> - Storage/Memory: The number of non-zero weights, which constitutes model storage, is identical by construction. The expanded network additionally stores which sub-neuron each weight belongs to, but this requires only log₂α bits per weight, negligible relative to the weight values. The network does use more activation memory during forward passes, but this overhead is modest relative to the weight storage.
> - Compute: Since the sparse mask is fixed after initialization and only modest fine-tuning is required after splitting, training overhead is likely limited. For inference, we provide initial evidence of hardware compatibility. Table A12 shows that FPE with 2:4 structured sparsity, natively supported by NVIDIA Ampere architecture (Pool et al., 2021), achieves performance comparable to unstructured FPE. Recent work on sparse matrix multiplication (e.g., Macko & Boža, 2025) further suggests that practical efficiency of sparsification is rapidly improving. We acknowledge that full wall-clock benchmarking is needed, which we will discuss in the limitations, but view this work as demonstrating the phenomenon exists, for future practical improvements.
> ### 3. Parameter count after splitting
> We appreciate the opportunity to clarify, as this is central to FPE. Here is a concrete example: a single neuron has 8 incoming weights, yielding 8 non-zero parameters. We split this neuron into 2 sub-neurons, each receiving half of the  inputs, so each has 4 non-zero weights. The total number of non-zero parameters is: 2 sub-neurons × 4 non-zero weights = 8 non-zero parameters, identical to the original. We now have more neurons, but each is sparser, and the total non-zero weight count is preserved, thus changing the allocation of parameters without changing the parameter budget. This is formalized as Algorithm A1 in the appendix. We will make this more clear in the writing as well.
> ## Response to Weaknesses
> ### 1. Small-scale experiments:
> The focus on smaller, controlled settings is intentional. We aim to fully characterize and interpret the interference-reduction mechanism in regimes where such analysis is tractable, analogous to how the lottery ticket hypothesis (Frankle and Carbin, 2018) and the superposition hypothesis (Elhage et al., 2022) were established in small models before being studied at scale. That said, we do provide evidence of scaling along multiple axes, such as in deeper architectures (5, 7, and 9-layer MLPs in Table A11), a large output space (ImageNet-1k in Table A6), and joint feature learning with CNN backbones (appendix A.7.1). Extension to transformers and large-scale pretraining is an exciting direction for future work.
> ### 2. Writing clarity:
> We appreciate this feedback. In the revised version, we will add a clearly structured list of contributions and an outline paragraph in the introduction to improve navigability, and we will revise the writing to make the presentation less dense.
>
> Thank you once again for your insightful comments and questions, which helped us to clarify key points of our work in the revision.
>
> Kaplan et al. 2020: https://arxiv.org/abs/2001.08361
>
> Macko and Boža 2025: https://arxiv.org/abs/2511.13061

---

> > ### Author Rebuttal · Reviewer_GceV · 2026-04-02
> >
> > Thank you for your detailed response. You have addressed my key questions and concerns, and I am satisfied with the clarifications. I will maintain my positive score.

---

> > > ### Author Response · Authors · 2026-04-05
> > >
> > > Thank you for your follow up and for maintaining your positive assessment! We are glad that our response addressed your key questions and concerns, and we will incorporate these clarifications in the revision to further strengthen the paper.

---

### Decision · Program_Chairs · 2026-04-30

**Decision:**

Accept (regular)

**Comment:**

This paper shows that increasing the number of neurons in a neural network while maintaining the number of non-zero parameters, by splitting neurons and then pruning, can significantly improve model performance. The authors attribute this to reduction in interference between features in a polysemantic neuron, and motivate this through learning boolean DNFs. They do preliminary experiments on more realistic datasets like MNIST, CIFAR and Imagenet 1k.

The reviewers found the paper technically sound and fairly original, with a clear and interesting idea to reduce feature interference. The main weaknesses were primarily about framing and scope. In particular, (1) the contributions and novelty relative to prior sparse-training work were not clear, (2) there was insufficient discussion of when the method would be most helpful and when not, and (3) while the results were compelling in the toy setting, they were only suggestive on real data. The reviewers also found the paper hard to read. Despite these weaknesses, the reviewers were overall positive, and happy to accept the paper as long as the paper's claims are clearly and honestly presented and the exposition significantly improved. Therefore I recommend acceptance and encourage the authors to take the feedback strongly and submit an improved camera-ready version.